# ScaleEnv: Scaling Environment Synthesis from Scratch for Generalist Interactive Tool-Use Agent Training

**Dunwei Tu** [* 1 2 †]  **Hongyan Hao** [* 3]  **Hansi Yang** [* 3]  **Yihao Chen** [4 †]  **Yu Yang** [3]  **Yueqing Sun** [3]  **Xingchen Liu** [5 †]
**Furao Shen** [1 2]  **Qi Gu** [3]  **Hui Su** [3]  **Xunliang Cai** [3]

## Abstract

Training generalist agents capable of adapting to diverse scenarios requires interactive environments for self-exploration. However, interactive environments remain critically scarce, and existing synthesis methods suffer from significant limitations regarding environmental diversity and scalability. To address these challenges, we introduce ScaleEnv, a framework that constructs fully interactive environments and verifiable tasks entirely from scratch. Specifically, ScaleEnv ensures environment reliability through procedural testing, and guarantees task completeness and solvability via tool dependency graph expansion and executable action verification. By enabling agents to learn through exploration within ScaleEnv, we demonstrate significant performance improvements on unseen, multi-turn tool-use benchmarks such as $\tau^2$-Bench and VitaBench, highlighting strong generalization capabilities. Furthermore, we investigate the relationship between increasing number of domains and model generalization performance, providing empirical evidence that scaling environmental diversity is critical for robust agent learning.

## 1. Introduction

The rapid evolution of Large Language Models (LLMs) has established a foundation for Artificial General Intelligence (AGI), driven largely by the success of Data Scaling

---
[*]Equal contribution †Work done during an internship at Meituan [1]National Key Laboratory for Novel Software Technology, Nanjing University [2]School of Artificial Intelligence, Nanjing University, Nanjing, China [3]Meituan, Beijing, China [4]Institute of Computer Science and Technology, Harbin Institute of Technology (Shenzhen), Shenzhen, China [5]School of Statistics, East China Normal University, Shanghai, China. Correspondence to: Hongyan Hao <haohongyan02@meituan.com>, Furao Shen <frshen@nju.edu.com>, Qi Gu <guqi03@meituan.com>.

*Proceedings of the $43^{rd}$ International Conference on Machine Learning*, Seoul, South Korea. PMLR 306, 2026. Copyright 2026 by the author(s).

and Model Parameter Scaling laws (Kaplan et al., 2020; Grattafiori et al., 2024; OpenAI, 2025; Team et al., 2026). However, transforming these models from text generators into agents requires a fundamental shift: agents must effectively interact with dynamic environments and iteratively refine their actions based on environment feedback (Yao et al., 2022; Jin et al., 2025; Zhang et al., 2024; Lu et al., 2025). Achieving such capabilities calls for training LLMs within dynamic environments, equipped with executable tools to allow agents to interact and receive immediate feedback. However, constructing environments that can be used for training LLM agents presents two core challenges. The first is *Realism*: tools synthesized directly by LLMs are often functionally unreliable, while LLM-based simulators are prone to severe hallucinations (Liu et al., 2024; Li et al., 2025). We argue that environments must be grounded in verified, executable code rather than probabilistic text generation to ensure robust feedback. The second challenge is *Scalability*: synthesis cannot rely on finite external documentation or manual human intervention (Cai et al., 2025).

To overcome these challenges, we introduce SCALEENV, a framework enabling the fully automated construction of high-fidelity, interactive environments and verifiable tasks. SCALEENV operates through two synergistic phases that ensure both code-level rigor and real-world complexity. In the first phase, the system builds the *Domain Foundation* by leveraging LLMs to define tool and database schemas from simple domain keywords. It employs a multi-agent architecture to generate functional code for tools and databases, which is rigorously validated via a *Procedural Testing* mechanism to guarantee error-free execution. These components are then consolidated into a global *Tool Dependency Graph* to map logical relationships. In the second phase, we focus on *Task Construction* using a dependency-aware expansion strategy. By sampling seed tool chains from the graph and dynamically introducing associated database states and distractor data, the system "snowballs" the environment state from linear paths into complex non-linear subgraphs. This process ensures the synthesized environment supports open-ended exploration and diverse solution paths, providing a verifiable foundation for grounding natural language user intents.

SCALEENV produces a complete training ecosystem com-

prising executable toolkits, high-fidelity environment states, and verifiable user intents. By training Qwen-3 models (Yang et al., 2025) via Zero Reinforcement Learning (RL) on this synthesized universe, we observe substantial performance boosts on unseen Out-of-Distribution (OOD) benchmarks, including $\tau^2$-Bench (Barres et al., 2025) and VitaBench (He et al., 2025). Crucially, this evaluation is strictly OOD: the synthesized training domains are entirely disjoint from the evaluation domains, and the benchmarks present distinct data formats (e.g., policy-constrained dialogues) not encountered during training. Furthermore, we empirically characterize the impact of Environment Scaling on model generalization performance, revealing a distinct curve that validates environmental diversity as a critical determinant for robust generalization.

Our main contributions are as follows:

- We propose SCALEENV, a fully automated framework that synthesizes high-fidelity interactive environments from scratch. It establishes an large-grade pipeline for agentic data, circumventing the limitations of fixed environment and manual API integration.
- We design a robust synthesis mechanism combining *Procedural Testing* and *Graph Expansion*. This approach ensures that generated environments possess rigorous code-level verifiability while maintaining the logical complexity required for deep reasoning.
- We demonstrate that agents trained on SCALEENV achieve significant zero-shot generalization on unseen benchmarks. Additionally, we provide empirical evidence for the *Environment Scaling Curve*, establishing a new paradigm for data-centric agent training.

## 2. Related Works

### 2.1. Tool Learning

The field of tool learning has evolved from Supervised Fine-Tuning (SFT) to autonomous exploration. Early works (Schick et al., 2023; Qin et al., 2023; Liu et al., 2024; Prabhakar et al., 2025) demonstrated that LLMs could master function calls through static demonstrations. To reduce dependency on expensive expert trajectories and enhance agents' self-exploration capabilities, frontier research has pivoted towards RL (Luo et al., 2025; Jin et al., 2025; Lu et al., 2025). However, large-scale RL exploration necessitates scalable interaction environments. This paper aims to bridge this gap by synthesizing diverse and reliable environments, enabling the training of highly generalizable tool-use agents.

### 2.2. Environment Scaling

Constructing effective environments for agents requires considering three critical dimensions: diversity, realism, and scalability. Existing approaches fall into three categories.

(1) **Real-world environments.** (Fang et al., 2025; Xu et al., 2025; Yao et al., 2026) collect actual tools or remote APIs. Although offering high realism, they are constrained by limited domain availability and safety policies that restrict action spaces. Consequently, the lack of diverse state-altering tasks, combined with prohibitive latency and costs, creates a bottleneck for scalable agent training. (2) **LLM-simulated environments.** (Liu et al., 2024; Chen et al., 2025; Li et al., 2025; Team et al., 2025; Ye et al., 2025) leverage large language models to generate tool responses and execution results, offering significant advantages in scalability, low-cost execution, and flexible domain definition. However, these approaches often suffer from fundamental limitations in realism and fidelity; specifically, they are prone to hallucinations and frequently fail to maintain authentic environment states. (3) **Synthetic environments.** While recent frameworks like AutoForge (Cai et al., 2025) and EnvScaler (Song et al., 2026) offer executable pipelines for environment synthesis, they face distinct limitations: the former is constrained by the limited scalability of document-based generation, while the latter struggles to construct complex, user-interactive tasks in RL stage. Furthermore, both approaches exhibit inadequate consistency between the generated tasks and their corresponding environmental states, undermining the reliability of the resulting sandboxes. Critically, existing methods (Liu et al., 2024; Li et al., 2025; Song et al., 2026) rely heavily on expensive proprietary teacher models for data generation, and depend on rigid predefined task sets or specific benchmarks as priors, limiting their scalability and generalizability. To address these defects, we introduce SCALEENV, which ensures coherence and execution reliability through execution-based verification. Unlike prior approaches, SCALEENV completely eliminates teacher model dependency by leveraging pure RL self-exploration, and requires only lightweight domain keywords as input, making it inherently more scalable. By providing a high-fidelity sandbox, SCALEENV enables the robust and scalable RL necessary for complex reasoning tasks.

## 3. Preliminaries

To construct a task for agentic RL training, we first need a domain foundation $\mathcal{B} = \langle \Sigma, \mathbb{T} \rangle$. It consists of two parts: a set of databases $\Sigma$ and a set of executable tools $\mathbb{T}$. $\Sigma$ defines the valid space of environment state $\mathcal{S}_{valid}^{env} = \{ s^{env} \mid s^{env} \models \Sigma \}$, while $\mathbb{T}$ contains all available functions or APIs in this domain.

Following the domain foundation $\mathcal{B}$, we can construct an environment $\mathcal{E} = \langle s_0^{env}, \mathbb{T} \rangle$ consisting of a set of databases $s_0^{env}$ with values filled following the database schema in $\mathcal{B}$ and executable tools $\mathbb{T}$ inherited from $\mathcal{B}$. $\mathcal{E}$ represents the external world at the start of an episode, as $s_0^{env}$ directly gives the initial hidden external environment state.

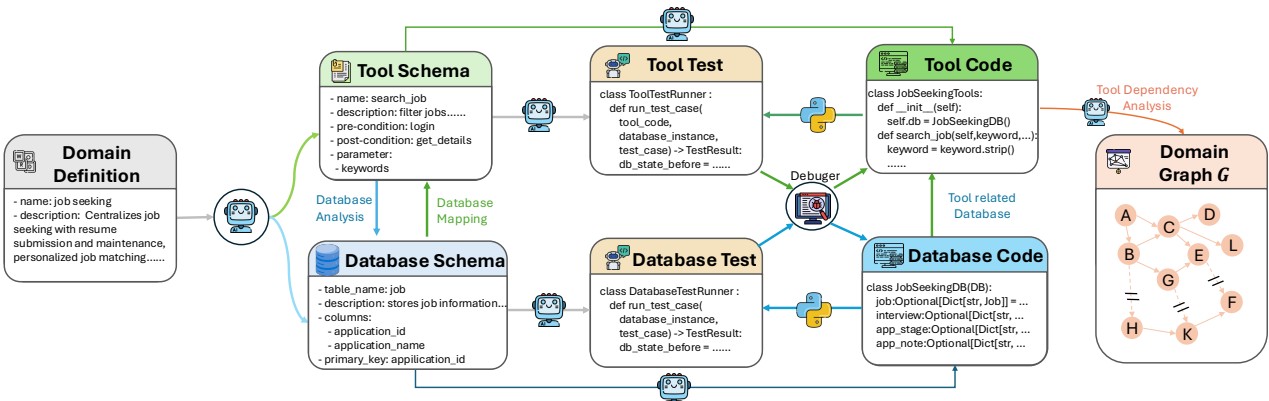

*Figure 1.* **Overview of Executable Graph Construction.** The pipeline proceeds from left to right: (1) Schema Definition for tools and databases; (2) Implementation validated via procedural testing; and (3) Tool Dependency Graph Construction to model execution logic.

Under a specific environment $\mathcal{E}$, a task $\psi = \langle \mathcal{E}, u, P_{user} \rangle$ then binds it to a specific user with some hidden goals, denoted as $u$, as well as the user profile $P_{user}$ that contains all necessary information about the user (e.g., permissions, location, history). The interaction between an LLM agent and the simulated user to accomplish this task can then be formulated as as a Partially Observable Markov Decision Process (POMDP) $\mathcal{M} = \langle \mathcal{S}, \mathcal{A}, \mathcal{O}, \mathcal{T}, \mathcal{R} \rangle$. Each state $s_t = (s_t^{env}, h_t, u) \in \mathcal{S}$ consists of three components: current environment state $s_t^{env}$, the interaction history $h_t$ and the user intent $u$. The action space $\mathcal{A} = \mathcal{A}_{resp} \cup \mathcal{A}_{tool}$ is defined as the union of natural language responses space $\mathcal{A}_{resp}$, and the space of tool execution commands $\mathcal{A}_{tool}$ for tools in $\mathbb{T}$. Correspondingly, the observation space $\mathcal{O} = \mathcal{O}_{resp} \cup \mathcal{O}_{tool}$ comprises user feedback space $\mathcal{O}_{resp}$ and tool execution results space $\mathcal{O}_{tool}$. The state transition function $\mathcal{T} : \mathcal{S} \times \mathcal{A} \rightarrow \mathcal{S} \times \mathcal{O}$ evolves the state based on the action type: an action $a_t \in \mathcal{A}_{tool}$ triggers a deterministic update to both the environment $s_t^{env}$ and the history $h_t$ with the return value of tools, whereas an action $a_t \in \mathcal{A}_{resp}$ updates only the history $h_t$ with the simulated user's reply, leaving $s_t^{env}$ unchanged. Finally, suppose the interaction is terminated at timestep $T$, the outcome reward signal is defined as $r = \mathcal{R}(s_T^{env}, u)$, which evaluates whether the final environment state $s_T^{env}$ successfully satisfy the user intent $u$.

# 4. Method

Motivated by the limitations of existing works that fail to synthesize diverse and reliable environments for agentic RL training, we present SCALEENV, a unified framework designed to synthesize multi-turn, interactive, and strictly verifiable environments, which can effectively scale up agentic RL training along with their corresponding tasks. To ensure modularity and extensibility, we decouple the environment and task construction into two distinct phases: *Executable Graph Construction* (Section 4.1) and *Task Instantiation*

(Section 4.2). The executable graph establishes the logical skeleton of a given domain. Defined as a graph comprising a set of executable tools, it determines the available action space and the dependency relationships within the domain. Following the construction of executable graph, we can further generate a diverse set of high-quality tasks for each domain. Ultimately, these synthesized tasks serve as the foundation for scalable agentic RL training.

## 4.1. Executable Graph Construction

### 4.1.1. TOOL & DATABASE SCHEMA DEFINITION

To establish a robust interactive domain foundation, we introduce a two-step synthesis pipeline. As shown in Figure 1, we first rigorously define *Tool & Database Schemas* to formalize the domain's operational logic, and subsequently implement the corresponding *Executable Code* to transform these abstract definitions into a functional, verifiable sandbox.

**Top-Down Tool Schema Synthesis.** Starting with a specific domain name (e.g., "Job Seeking"), we employ a top-down synthesis approach by using an LLM to first conceptualize the domain logic and generate the *Tool Schema*. This schema rigorously defines the interface of the atomic tool set $\mathbb{T}$, including precise functional descriptions, parameters, and logical pre/post-conditions (e.g., `submit_application` logically necessitates a preceding `upload_resume`).

**Database Schema Derivation & Mapping.** With the tool schema synthesized, a Database Agent analyzes the tool definitions to reverse-engineer the database structure necessary to support the environment. For instance, the presence of a `submit_application` tool implies the existence of an `Application` table (and a reference `Job` table) in the environment database. Through de-duplication and filtering, the agent derives a consolidated *Database Schema*

for $\mathcal{S}^{env}_{valid}$, defining table structures and integrity constraints. We simultaneously establish a *tool-database mapping* that explicitly identifies the specific tables associated with each tool, laying the necessary groundwork for the subsequent code implementation.

**Reward Specification** While the LLM-as-a-judge paradigm is widely adopted to implement the reward function $\mathcal{R}$, it often suffers from high computational overhead and vulnerability to *reward hacking* (Gabor et al., 2025; Pan et al., 2024). As such, we introduce a rule-based evaluator that directly checks the agent's final database state $s^{env}_T$ against the ground-truth state $s^{env}_{gt}$. Note that different types of data naturally require different criteria ( e.g., critical data like prices or quantities require exact match, while text comments may only require fuzzy match), we categorize database columns into three matching policies: (1) **Exempt Fields**: dynamically generated IDs and optional columns that do not affect task success; (2) **Hard Constraints**: critical data such as timestamps or quantities that require strict character-level or numerical equality; and (3) **Semantic Alignment**: descriptive text that only require fuzzy semantic matching.

### 4.1.2. TOOL & DATABASE SCHEMA IMPLEMENTATION

Given the synthesized tool and database schema, we then proceed to implementing them through LLM-powered code generation. Since the database defines the underlying state structure required for tool execution, we prioritize generating and verifying the database code, which then serves as the necessary prerequisite for the subsequent tool implementation and verification.

**Database Implementation & Verification.** We first utilize an LLM to translate the database schema into executable code. To ensure reliability, we concurrently generate test scripts to validate the database implementation against integrity constraints. Any execution failures trigger another LLM agent that works as a debuger and iteratively refines the code based on error tracebacks until all tests pass, establishing a stable storage layer for the environment.

**Tool Implementation via Procedural Testing.** Note that generating valid tool code is a non-trivial process involving intricate logic and interactions across multiple databases, and direct generation is prone to hallucination. As such, we propose a *Procedural Testing* mechanism. Specifically, the Code Agent implements the tool logic, while a Test Agent simultaneously synthesizes unit test cases and corresponding *matched database instances*. We then execute the tool code on the matched database instances and validate its correctness based on three distinct outcomes:

- **Success:** The execution completes without error, and the resulting state transitions strictly match the expected

database states.

- **Anticipated Rejection:** The tool correctly identifies and handles invalid inputs by raising the pre-defined exceptions as specified in the schema.

- **Unexpected Failure:** Any other runtime errors or state inconsistencies indicate defects. In this case, the Debug Agent analyzes the error logs to iteratively rectify either the tool implementation or the database instance until the procedural test is satisfied.

### 4.1.3. TOOL DEPENDENCY GRAPH CONSTRUCTION

To facilitate the synthesis of semantically coherent multi-step tasks, we further employ a *Tool Dependency Agent* to systematically evaluate pairwise relationships between the verified tools. This analysis is grounded in three key dimensions: *data flow* (parameter passing), *pre/post-conditions* (logical prerequisites), and *state dependencies* (shared database tables). Based on these criteria, the agent establishes directed edges representing causal links, consolidating the atomic tools into a unified *Tool Dependency Graph G*. This graph serves as the basis for subsequent task instantiation.

### 4.2. Task Instantiation via Graph Expansion

Building upon the tools and databases constructed in Section 4.1, we instantiate diverse tasks for agentic RL training. The primary challenge for task instantiation is to construct a high-fidelity environment $\mathcal{E}$ capable of supporting the extensive trial-and-error in RL. Unlike SFT where no trial-and-error mechanism exists, an RL environment must satisfy two critical requirements:

- **Entity Consistency.** The synthesized environment should be consistent across all database tables. An entity appearing in one table (e.g., a user_id in the *Order* table) must map correctly to corresponding entities in related tables (e.g., the *User* table).

- **Interaction Completeness.** The environment must support *execution fidelity* across the entire feasible action space, not merely along the optimal trajectory. Formally, for any valid tool calling action $a \in \mathcal{A}_{tool}$ taken by the agent, the environment $\mathcal{E}$ must return a valid, semantically meaningful observation $o_{tool}$, ensuring that exploration is not artificially terminated due to missing database entries or implementation gaps.

To generate tasks with environments satisfy the two constraints above, we propose a *Graph Expansion* strategy. As illustrated in Figure 2, this process constructs complex environmental states via a two-stage iterative procedure:

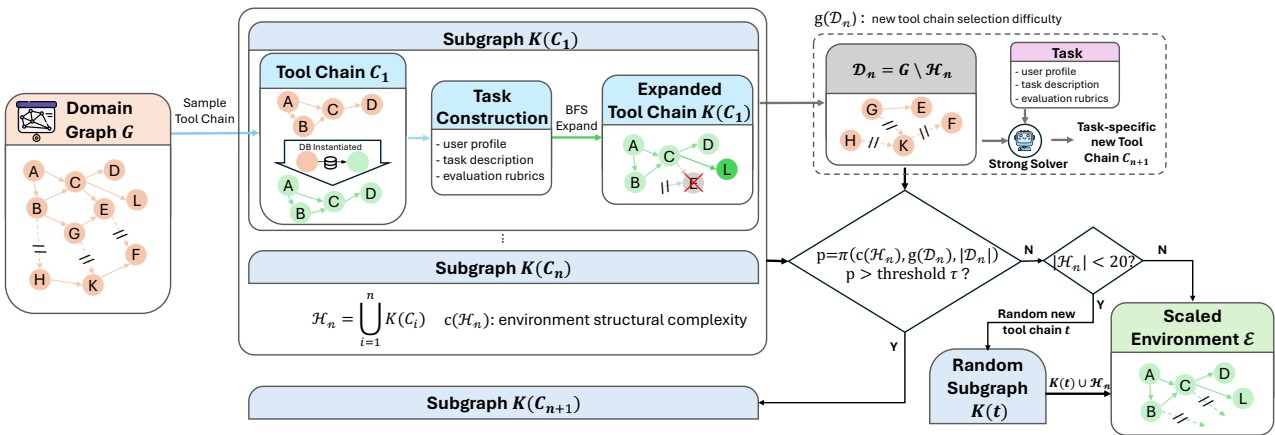

*Figure 2.* **Overall pipeline of Task Instantiation via Graph Expansion.** The process involves: (1) Seed Chain Sampling from the dependency graph; (2) Task Initialization with verifiable execution; and (3) Controlled Environment Expansion to scale complexity while maintaining solvability.

*seed tool chain sampling*, which initializes a seed subgraph $K(C_1)$, followed by *controlled environment expansion*, which expands the environment based on complexity metrics and scales via a fallback mechanism.

#### 4.2.1. TASK INITIALIZATION WITH SEED TOOL CHAIN

Given the domain-specific tool dependency graph $G$, we initialize the complete task construction process through three stages: (1) executable seed tool chain sampling, (2) constraint-satisfying environment instantiation and (3) grounded instruction synthesis.

**Executable Seed Tool Chain Sampling.** We initiate the process by sampling a *seed tool chain* $C_1 = (a_1, a_2, \ldots, a_k)$, which serves as a valid reference path to solve the task to be constructed. The tool chain is formulated as an executable code, and generated by prompting an LLM with the tool dependency graph $G$ and the relevant database schema. This setup enables the joint modeling of tool sequences and their arguments, preventing possible disconnection if parameter instantiation is treated as an isolated step. Representing the tool chain as code also inherently satisfies data flow constraints; the output of a preceding action $a_i$ is programmatically propagated as the input to subsequent actions $a_{i+1}$.

**Initial State Construction with Distractor Injection.** Based on the generated tool chain $C_1$, we construct an initial environment state $s_0^{env}$ that strictly supports its execution while preserving *Entity Consistency*. We employ an LLM-based generation pipeline to synthesize $s_0^{env}$ and validate it by executing $C_1$ on it to ensure task feasibility. Furthermore, to encourage robust reasoning, we populate the database tables within $s_0^{env}$ with additional records that act as *dis-*

*tractors.* The density of these distractors is dynamically scaled according to the predefined task complexity. While these distractors adhere to all schema constraints, they remain functionally orthogonal to the ground-truth trajectory, forcing the agent to acquire precise information filtering capabilities.

**Instruction Synthesis.** Given the verified seed chain $C_1$ and environment state $s_0^{env}$, we then employ an LLM to synthesize the user profile $P_{user}$ and user instruction $u$. We ensure the generation of $u$ to be strictly grounded in $C_1$ as the reference solution, which prevents the introduction of external priors or hallucinations unsupported by the underlying environment. The evaluation criteria $\mathcal{R}$ is also directly derived from the final state $s_{gt}^{env}$ after executing $C_1$, which ensures alignment between the seed tool chain $C_1$ and reward $\mathcal{R}$ for robust RL training.

#### 4.2.2. CONTROLLED ENVIRONMENT EXPANSION

While the three-stage procedure in Section 4.2.1 ensures *Entity Consistency*, restricting the agent to the minimal environment state encourages overfitting, where the agent collapses into memorizing sparse trajectories rather than learning generalized reasoning. To foster robust RL training and ensure *Interaction Completeness*, we propose an iterative environment refinement strategy. We first expand the initial seed chain $C_1$ into a semantically dense subgraph $\mathcal{H}_1 = K(C_1) \subset G$ and incrementally refine the environment. Then we further try to construct more tool chains $C_2, \ldots, C_n$ until we reach the capability ceiling of current LLM, and repeat the expansion-refinement process of $C_1$ to these newly constructed tool chains $C_2, \ldots, C_n$ for further environment refinement.

**Dependency-Aware Topological Expansion.** We first expand the initial tool chain $C_1$ into a local subgraph $\mathcal{H}_1 = K(C_1) \subset G$ for environment refinement. A naive stochastic injection of tools risks introducing *dependency dead-ends*—nodes whose prerequisite inputs cannot be satisfied by the current available tool output. We mitigate this via *Dependency-Aware BFS*: starting with $\mathcal{H}_1 = C_1$, we iteratively traverse $G$ and add a new tool node $v \in G$ to $\mathcal{H}_1$ if and only if the input and output dependencies of $v$ can be fully satisfiable by a tool subset of $\mathcal{H}_1$. Then for any newly added tool node $v$, we further execute it with its argument derived from $\mathcal{H}_1$ and refine the environment if any errors arise from the execution.

**LLM-Gated Chain Expansion.** Since the subgraph $\mathcal{H}_1$ is derived from the only seed tool chain $C_1$, a natural idea is to extend similar procedure to multiple tool chains $C_1, C_2, \ldots, C_n$. Nevertheless, such extension cannot be done forever as the new tool chain should only contain tools not used in previous subgraphs $\mathcal{H}_n = \bigcup_{i=1}^{n} K(C_i)$, As such, letting $\mathcal{D}_n = G \setminus \mathcal{H}_n$ denote the set of candidate tools available after iteration $n$, we determine whether to inject a new seed chain $C_{n+1} \subseteq \mathcal{D}_n$ via a parametric gating policy $\pi$. Instead of relying on brittle heuristics, we employ a strong LLM to approximate a value function for expansion, conditioning on the following metrics:

- **Structural Complexity** ($c(\mathcal{H}_n)$). To quantify the representational sufficiency of tools currently in $\mathcal{H}_n$, we introduce a complexity metric $c(\mathcal{H}_n) = \frac{|V_{\mathcal{H}_n}| + \lambda |E_{\mathcal{H}_n}|}{S_{\text{sat}}}$, where $V_{\mathcal{H}_n}$ and $E_{\mathcal{H}_n}$ represent the number of nodes and edges in $\mathcal{H}_n$, $\lambda = 0.5$ weighs the complexity of dependencies, and $S_{\text{sat}} = 50$ is the saturation constant. Intuitively, this score assesses whether $\mathcal{H}_n$ captures the necessary structural and semantic depth required to support complex environmental interactions.

- **Feasibility Score** ($g(\mathcal{D}_n)$). Complementary to the tool complexity, we also need to consider the feasibility of the task within the remaining tools in $\mathcal{D}_n$. Since ground truth labels for complex tool chains are often unavailable in open-ended environments, we rely on a powerful "Oracle" agent and define $g(\mathcal{D}_n) \in [0, 1]$ as the success rate of this oracle agent in identifying executable tool chains within $\mathcal{D}_n$. In our implementation, we instantiate use Qwen3-235B-A22B augmented with a best-of-$k$ search strategy ($k = 16$ rollouts) to maximize the likelihood of discovering valid paths.

Along with the number of available tools $|\mathcal{D}_n|$, we construct a prompt $\pi(|\mathcal{D}_n|, c(\mathcal{H}_n), g(\mathcal{D}_n))$ instructing the LLM to balance the trade-off between diversity and solvability. The model outputs a compatibility score $p \in [0, 1]$. If $p \geq \tau$, we sample a new chain $C_{n+1}$, expand it into its dependency

subgraph $K(\mathcal{C}_{n+1})$ following similar procedure that obtains $K(C_1)$ from $C_1$. We then also execute tools in $K(\mathcal{C}_{n+1})$ and refine the environment if any errors arise from the execution, before merging it into $\mathcal{H}_n$ as $\mathcal{H}_{n+1} = \mathcal{H}_n \cup K(\mathcal{C}_{n+1})$.

Finally, to ensure sufficient exploration space, we enforce a minimum constraint $|\mathcal{H}_n| \geq 20$ for $\mathcal{H}_n$. If it falls short, we randomly sample valid auxiliary chains and merge it into $\mathcal{H}_n$ to satisfy this constraint. This strategy leverages the semantic reasoning of LLMs to dynamically balance the environment construction, producing high-fidelity tasks that support complex reasoning while maintaining verifiable supervision signals.

## 5. Experiments

### 5.1. Experimental Setup

For the domain foundation and task synthesis phases, we utilized a diverse suite of high-performance LLMs, including Deepseek-V3.2 (Liu et al., 2025), GLM-4.7 (Zeng et al., 2025), GPT-5.1, and Qwen3-32B (Yang et al., 2025), to instantiate various agent roles. Across the 16 synthesized domains, each environment comprises 20–80 tools and 5–25 database tables. Our model series, denoted as **Qwen3-SE** (**S**cale**E**nv), is trained from Qwen3 (Yang et al., 2025) using group relative policy optimization (GRPO) (Shao et al., 2024) on our synthesized domains and tasks. We use **Qwen2.5-72B-Instruct** (Qwen et al., 2025) as the user simulator to provide natural language feedback. Regarding hyperparameters, the **Qwen3-8B** model was trained with a rollout batch size of 1024, while the **Qwen3-32B** model utilized a rollout batch size of 2048. Qwen3-32B was trained for 48 steps and Qwen3-8B was trained for 80 steps with a learning rate of $10^{-6}$. Detailed domain and task compositions are provided in Appendix B.

### 5.2. Main Results: Generalization to Unseen Domains

A critical question in agentic training is whether performance gains stem from genuine reasoning capabilities or merely from overfitting to the training distribution. To answer this question, we assess the generalization capabilities of **Qwen3-SE** model series across three dimensions using established benchmarks. (1) *Reasoning Generalization*: We utilize the cross-domain subset of VitaBench (He et al., 2025), which presents ambiguous user needs requiring proactive information retrieval and complex multi-step planning, testing the transfer of reasoning skills to challenging logical structures not seen during training. (2) *Domain Generalization*: We evaluate the model performance across a wide spectrum of functional areas, including the Airline, Retail, and Telecom domains from $\tau^2$-Bench (Barres et al., 2025), and the Delivery, In-store, and OTA domains from VitaBench. As visually evidenced by the tool

*Table 1.* **Zero-shot generalization performance.** The **Qwen3-SE** model series, trained with environments and tasks constructed from our SCALEENV consistently outperforms baselines across diverse domains.

| Model | $\tau^2$-Bench | | | VitaBench | | | |
|---|---|---|---|---|---|---|---|
| | **Retail** | **Airline** | **Telecom** | **Cross** | **Delivery** | **Instore** | **OTA** |
| *Open-weights Models* | | | | | | | |
| GPT-OSS-120B-A5B | 57.0 | 38.0 | 45.6 | 15.0 | 37.0 | 42.0 | 12.0 |
| Qwen3-235B-A22B-2507 | 71.9 | 58.6 | 47.3 | 14.5 | 45.0 | 32.0 | 15.8 |
| Kimi-K2-0905 | 70.6 | 56.5 | 65.8 | 11.5 | 32.5 | 30.0 | 18.8 |
| Seed-OSS-36B | 68.4 | 52.0 | 41.2 | 6.1 | 26.0 | 39.0 | 7.0 |
| xLAM-2-32B-fc-r | 55.3 | 52.0 | 16.7 | 4.0 | 26.0 | 17.0 | 10.0 |
| *LLM-based Environment Construction Methods* | | | | | | | |
| TOOLACE (LLaMA-3.1-8B) | 38.7 | 18.0 | 21.2 | 0.0 | 3.3 | 0.3 | 0.3 |
| Simia (Qwen3-8B) | 52.9 | 40.5 | 15.6 | 0.0 | 2.3 | 0.3 | 0.8 |
| EnvScaler (Qwen3-8B) | 49.6 | 31.5 | 32.7 | 2.5 | 22.8 | 22.3 | 6.3 |
| *Our Results* | | | | | | | |
| Qwen3-8B | 38.4 | 30.5 | 21.5 | 1.5 | 18.3 | 14.8 | 4.5 |
| **Qwen3-SE-8B** | 50.9 (+12.5) | 37.5 (+7.0) | 27.2 (+5.7) | 3.0 (+1.5) | 26.3 (+8.0) | 23.8 (+9.0) | 7.0 (+2.5) |
| Qwen3-32B | 59.5 | 48.0 | 27.2 | 5.3 | 27.0 | 22.5 | 4.5 |
| **Qwen3-SE-32B** | 63.6 (+4.1) | 48.0 (+0.0) | 30.9 (+3.7) | 10.8 (+5.5) | 31.3 (+4.3) | 34.5 (+12.0) | 12.5 (+8.0) |

*Table 2.* Pass@4 on VitaBench which shows the model upper-bound performance.

| Model | Cross | Delivery | Instore | OTA | Avg. |
|---|---|---|---|---|---|
| Qwen3-8B | 6 | 43 | 44 | 15 | 27.0 |
| **Qwen3-SE-8B** | 12 | 56 | 57 | 18 | 35.8 |
| Qwen3-32B | 15 | 57 | 58 | 14 | 36.0 |
| **Qwen3-SE-32B** | 29 | 65 | 66 | 27 | 46.8 |

embedding distribution in Figure 4, these evaluation domains are semantically distinct and spatially separated from our 16 synthesized training domains, ensuring a rigorous Out-Of-Distribution (OOD) evaluation. (3) *Format Generalization*: While SCALEENV focuses on direct user-tool interaction without explicit policy documents, $\tau^2$-Bench requires agents to strictly adhere to lengthy textual policies during dialogue. Success here demonstrates that our agents can generalize their learned behavior to novel interaction formats and constraints not explicitly present during training.

**Reasoning Generalization.** A critical challenge in agentic tasks is interpreting ambiguous user needs that require multi-step reasoning and proactive planning. We observe substantial gains on VitaBench, a benchmark specifically designed to test these capabilities. For instance, when a user states "I am sick," the agent must infer a latent intent to "recommend light food". As shown in Table 1, **Qwen3-SE-32B** achieves a remarkable improvement in the most challenging cross-domain subset, doubling the performance of the base model. This indicates that training on SCALEENV's verifi-

able environments enables the agent to transfer high-level reasoning skills to diverse, logic-heavy scenarios.

**Domain and Format Generalization.** SCALEENV demonstrates robust transferability across entirely novel domains and interaction formats. (1) **Domain Adaptation**: Despite the strict exclusion of test domains from our training set, our method consistently boosts performance across all 7 evaluation domains. (2) **Format Adaptation**: Notably, $\tau^2$-Bench requires agents to strictly adhere to lengthy textual policies. While fundamentally differnet from SCALEENV, the consistent gains on $\tau^2$-Bench suggest that our agents have generalized the ability to follow constraints and handle complex state transitions, even when presented in a novel textual policy format.

**Performance Upper Bound Analysis.** Beyond improving average stability, SCALEENV significantly elevates the model's capability ceiling on complex tasks, as is illustrated by the Pass@4 score on VitaBench in Table 2 that measures the probability of generating at least one correct solution within four attempts. Notably, in the complex cross-domain subset, our method nearly doubles the success potential. This confirms that SCALEENV does not merely teach the model to memorize simple patterns, but fundamentally enhances its capacity to search for and execute correct solutions in more difficult tasks.

**Comparison with LLM-based Environment Construction.** As shown in Table 1, we further compare SCALEENV against three recent LLM-based environment construction baselines: TOOLACE (Liu et al., 2024), Simia (Li et al.,

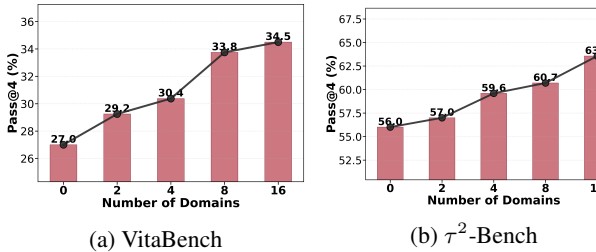

(a) VitaBench          (b) $\tau^2$-Bench

*Figure 3.* **Domain Scaling Analysis (Pass@4).** Comparison of zero-shot generalization as training domains scale from $N = 2$ to 16. $N = 0$ denotes the base model. Performance improves monotonically across both benchmarks.

2025), and EnvScaler (Song et al., 2026). While these methods achieve competitive performance on certain domains, they exhibit critical generalization weaknesses. Notably, Simia collapses to near-zero performance on VitaBench despite strong $\tau^2$-Bench results. This stems from its overfitting to user-driven session termination; when the agent must autonomously decide when to end the conversation (as in VitaBench), Simia endlessly repeats responses instead of concluding the dialogue. In contrast, SCALEENV consistently improves generalization across both interaction patterns without behavior collapse. Beyond performance, SCALEENV offers key methodological advantages: (1) it eliminates dependency on expensive proprietary teacher models (e.g., GPT-4o, GPT-5) by leveraging pure RL self-exploration, and (2) it requires only lightweight domain keywords as input, rather than rigid predefined task sets or benchmark-specific priors, making it inherently more scalable.

## 5.3. Domain Scaling Analysis

To investigate how environment diversity drives generalization, we evaluate Qwen3-8B trained on $N \in \{2, 4, 8, 16\}$ unique domains while keeping the same number of 1024 tasks. As illustrated in Figure 3, we observe a steady upward trend in zero-shot generalization as the number of training domains increases. This consistent growth across disparate benchmarks confirms that environmental richness is a decisive factor in unlocking the model's transfer capabilities. Furthermore, exposing agents to a wider variety of tools and databases allow them to internalize abstract, domain-agnostic reasoning strategies rather than overfitting to specific templates. Notably, performance has not yet fully plateaued at $N = 16$, suggesting that further scaling of environmental diversity remains a promising direction for data-centric agent training.

## 5.4. Analysis Experiment

**Ablation Study: Executability Verification.** A core tenet of SCALEENV is the rigorous *Execution-Based Verification* (EV) of synthesized tasks. To quantify its impact, we

*Table 3.* **Ablation of Executability Verification (Avg@4) on $\tau^2$-Bench**.

| Setting | Retail | Airline | Telecom |
|---|---|---|---|
| Qwen3-8B | 38.4 | 30.5 | 21.5 |
| w/o EV | 42.3 | 30.0 | 25.2 |
| **Qwen3-SE-8B** | **50.9** | **37.5** | **27.2** |

*Table 4.* **Ablation of Reward Mechanisms.** Results are averaged across three $\tau^2$-Bench domains (Retail, Airline, Telecom).

| Reward Setting | Avg@4 | Pass@4 | Pass^4 |
|---|---|---|---|
| LLM-as-a-Judge | 36.5 | 58.8 | 14.6 |
| **Rule-Based (Ours)** | **38.5** | **62.9** | **15.0** |

conducted an ablation study by training Qwen3-8B on a dataset generated without this verification step (denoted as "w/o EV"). In this setting, while the tool dependency graph was constructed, the tools are never subjected to actual parameter-driven execution, and the environment states are not iteratively patched to ensure solvability.

As illustrated in Table 3, the removal of execution verification leads to a consistent degradation in performance across all domains of $\tau^2$-Bench. Without EV, the training data contains tool calls that appear semantically plausible but fail during runtime due to unsatisfied preconditions or mismatched database states (e.g., attempting to refund a non-existent order). Such noisy rollouts introduce conflicting reward signals, preventing the policy from learning precise, logic-grounded decision-making.

**Ablation Study: Reward Mechanism.** To evaluate the impact of different reward signals, we compare our deterministic rule-based evaluator against the standard *LLM-as-a-Judge* paradigm on Qwen3-8B. As shown in Table 4, our rule-based approach yields superior performance across all metrics. These results demonstrate that while LLM-based judges provide semantic flexibility, they are often susceptible to *reward hacking*, where agents optimize for linguistic alignment at the expense of logical correctness. In contrast, our rule-based reward enforces rigorous, database-level fidelity, providing a more objective and robust learning signal. Furthermore, the rule-based approach minimizes computational overhead by replacing expensive LLM inference with efficient rule-based verification, facilitating stable large-scale RL exploration.

**Ablation Study: Component Contribution.** SCALEENV relies on two pillars: Procedural Testing (to ensure domain validity) and Graph Expansion (to ensure interaction completeness and entity consistency). To identify the component most responsible for performance gains, we conduct an ablation study on task construction success rate. As shown in Table 5, while removing Procedural Testing introduces

*Table 5.* **Ablation of core components on task construction success rate.**

| Model / Ablation | Success Rate (%) | Δ |
|---|---|---|
| SCALEENV (Full) | 66.2 | — |
| w/o Procedural Testing | 39.0 | −27.2 |
| w/o Graph Expansion | 27.0 | −39.2 |

*Table 6.* **Domain Stability Analysis (Avg@4).** Set A contains wedding planning, knowledge management, job seeking and healthcare telemedicine. Set B contains express logistics, job seeking, email management and pet care.

| Setting | Cross | Delivery | Instore | OTA | Avg. |
|---|---|---|---|---|---|
| Qwen3-8B | 1.5 | 18.3 | 14.8 | 4.5 | 9.8 |
| Set A (4 Domains) | **2.0** | 24.0 | **22.3** | 7.0 | **13.8** |
| Set B (4 Domains) | **2.0** | **25.5** | 18.3 | **7.5** | 13.3 |

faulty tools and significantly drops success rate, removing Graph Expansion fundamentally breaks the environment's internal consistency. Without it, the LLM hallucinates environment states, leading to broken tool-execution paths and the steepest drop in success rate (from 66.2% to 27.0%).

**Domain Stability Analysis.** A potential concern in procedural generation is whether performance gains stem from specific "lucky" domains or general synthesis robustness. To investigate this, we conducted a stability analysis by training the Qwen3-8B model on two distinct, non-overlapping subsets of synthesized dommains, fixing the domain count at 4 and task count at 1024. As reported in Table 6, both subsets consistently outperform the baseline across all metrics on VitaBench. This consistency demonstrates that EnvZero produces high-fidelity environments reliably across diverse scenarios, ensuring that generalization improvements are driven by the structural advantages of our framework rather than artifacts of specific cherry-picked domains.

**Cost Analysis.** SCALEENV is highly cost-effective and requires zero manual intervention. To quantify the generation cost, we randomly selected 4 domains (generating 20 tasks per domain) and tracked the resource consumption. As shown in Table 7, the schema definition stage achieves a 100% success rate with minimal retries, while the task completion stage requires slightly more retries due to the complexity of multi-step environment construction. Overall, the fully automated pipeline maintains high efficiency across all stages.

## 6. Conclusion

In this paper, we presented SCALEENV, a framework to synthesize high-fidelity interactive environments and verifiable tasks. By shifting from static dataset interpolation to a complete generation pipeline, SCALEENV circumvents the

*Table 7.* **Per-domain and per-task generation cost.** "Success Rate" denotes the pass rate at each stage before triggering an automated retry.

| Stage | Prompt Tokens | Completion Tokens | Success Rate (%) | Avg. Retries |
|---|---|---|---|---|
| Schema Def. | 26.3K | 31.3K | 100.0 | 1.00 |
| Schema Impl. | 202.2K | 64.7K | 89.8 | 1.27 |
| Task & Env Init. | 13.3K | 4.7K | 91.7 | 1.27 |
| Task & Env Comp. | 22.0K | 2.2K | 72.2 | 1.46 |

inherent limitations of data scarcity. Our extensive evaluations across multiple model scales demonstrate that training on SCALEENV-synthesized environments and tasks significantly boosts the performance of baseline models on unseen benchmarks, evidencing robust zero-shot generalization. Furthermore, we empirically formulated an domain scaling curve, establishing that scaling environmental diversity is more critical than task quantity for cultivating generalist agent capabilities. These results confirm that SCALEENV provides a stable and scalable paradigm for data-centric reinforcement learning, paving the way for the development of robust, general-purpose autonomous agents.

## Acknowledgements

This work was partially supported by the National Natural Science Foundation of China (Grant No. 62276127), Fundamental and Interdisciplinary Disciplines Breakthrough Plan of the Ministry of Education of China (No. JYB2025XDXM118), and the "111 Center" (No. B26023). This work was supported by the Meituan LongCat team. We also thank Zhikang Xia and Yi-Kai Zhang for their valuable contributions to this work.

## Impact Statement

This paper presents work whose goal is to advance the field of Machine Learning, specifically by addressing the data scarcity bottleneck in training generalist tool-use agents. Our proposed framework, SCALEENV, provides a safe, virtual sandbox for agents to learn complex tool-use capabilities, thereby mitigating the risks of accidental harm associated with training directly on real-world APIs.

However, we acknowledge the potential risks associated with the generative nature of our method. Theoretically, SCALEENV is domain-agnostic and capable of synthesizing arbitrary interactive environments. Without proper oversight, this capability could be misused to construct environments that model harmful or unethical behaviors. We strongly emphasize that the application of procedural environment synthesis must adhere to strict ethical guidelines. Future research should explore safety alignment mechanisms that prevent the synthesis of malicious domains while

maintaining the diversity required for robust generalist training.

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

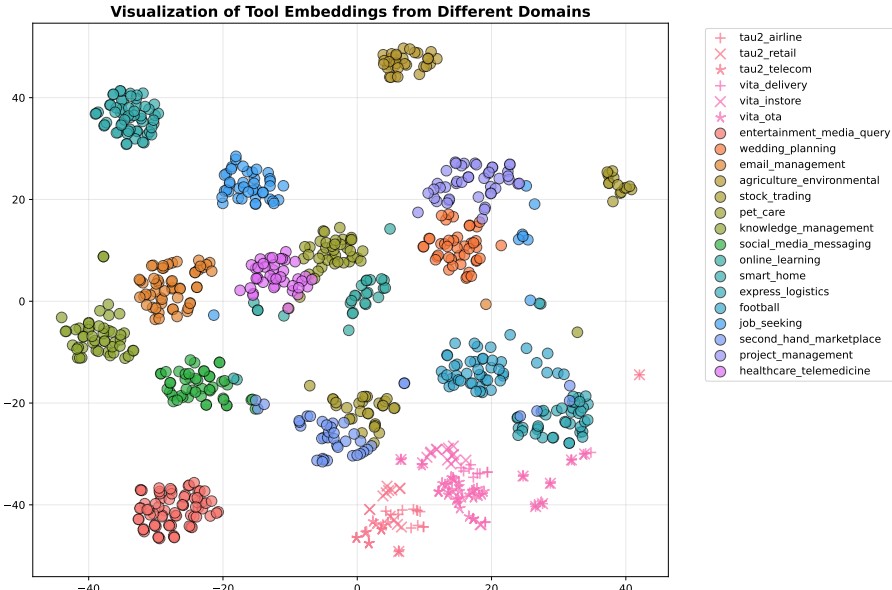

*Figure 4.* **Visualization of Tool Embeddings across Domains.** We use t-SNE ([Maaten & Hinton, 2008](#)) to project the semantic embeddings of tools from our 16 synthesized training domains (circles) and the evaluation benchmarks (crosses and pluses). The clear spatial separation between the training clusters and the $\tau^2$ / Vita domains empirically demonstrates the *OOD* nature of our evaluation.

*Table 8.* **Tool Schema Example (Domain: Job Seeking).** This table illustrates a subset of the tool action schema automatically generated by the meta-agent, detailing input requirements and output structures.

| Tool Name | Description | Input Parameters | Output Structure |
|---|---|---|---|
| `add_interview_feedback` | Records notes and performance ratings after an interview is completed. | `interview_id`* (str) `feedback_content`* (str) `performance_rating` (int) `created_at`* (str: YYYY-MM-DD) | Object containing the new `feedback_id` and confirmed `interview_id`. |
| `get_application_interviews` | Retrieves the complete schedule and details of interviews for a specific application. | `application_id`* (str) | List of interview objects (date, type, interviewer, location). |
| `search_apps_by_keyword` | Filters job applications based on keywords in the job title or company name. | `keyword`* (str) `search_fields` (List[str]) | List of matching application summaries and total match count. |
| `add_salary_expectation` | Updates a job application record with the user's expected salary range. | `application_id`* (str) `expected_salary_min`* (num) `expected_salary_max` (num) `salary_currency` (str) | Confirmation object with `application_id` and success boolean. |

## A. Tool Semantic Diversity and OOD Verification

To further investigate the structural diversity of the synthesized "domain universe", we visualize the semantic embeddings of all tools across both training and evaluation domains. We utilize a pre-trained encoder to extract tool descriptions' embeddings and project them into a 2D space using t-SNE.

As illustrated in Figure 4, our 16 synthesized domains (represented by colored circles) form widely distributed clusters, covering a broad semantic spectrum from *Smart Home* to *Healthcare Telemedicine*. Crucially, the evaluation domains from $\tau^2$-**Bench** and **VitaBench** (represented by cross and plus markers) are situated in distinct regions, exhibiting significant semantic separation from the training clusters.

This visualization provides empirical evidence for the **Out-Of-Distribution (OOD)** nature of our evaluation setup. It

*Table 9.* **Database Schema Example (Domain: Job Seeking).** This table illustrates a subset of the database schema automatically generated by the database agent. It defines entity structures, data types, and integrity constraints required to support the recruitment process tracking.

| Field Name | Data Type | Constraint / Description |
|---|---|---|
| *Table: Job_Application* (Core application records) | | |
| application_id | VARCHAR(10) | **Primary Key**. Unique identifier. |
| applicant_name | VARCHAR(100) | Full name of the applicant. |
| job_title | VARCHAR(200) | Title of the position applied for. |
| company_name | VARCHAR(200) | Name of the hiring company. |
| status | VARCHAR(50) | Default: submitted. e.g., under_review, rejected. |
| priority_level | INTEGER | Priority (1-5). Higher value indicates higher priority. |
| salary_currency | VARCHAR(10) | Default: USD. Currency for expectations. |
| created_at | DATETIME | Record creation timestamp. |
| *Table: Application_Stage* (Tracks hiring progression) | | |
| stage_id | VARCHAR(10) | **Primary Key**. Stage record identifier. |
| application_id | VARCHAR(10) | **Foreign Key** → job_application.application_id. |
| stage_name | VARCHAR(100) | e.g., phone_screening, technical_interview. |
| stage_date | DATETIME | Timestamp when the stage was reached. |
| stage_notes | TEXT | Optional notes regarding this stage. |
| *Table: Interview_Schedule* (Manages interview logistics) | | |
| interview_id | VARCHAR(10) | **Primary Key**. Schedule identifier. |
| application_id | VARCHAR(10) | **Foreign Key** → job_application.application_id. |
| interview_type | VARCHAR(50) | e.g., behavioral, system_design. |
| interview_date | DATETIME | Scheduled date and time. |
| interviewer_name | VARCHAR(100) | Name of the assigned interviewer. |
| *Table: Interview_Feedback* (Post-interview evaluation) | | |
| feedback_id | VARCHAR(10) | **Primary Key**. Feedback identifier. |
| interview_id | VARCHAR(10) | **Foreign Key** → interview_schedule.interview_id. |
| feedback_content | TEXT | Detailed notes or feedback content. |
| performance_rating | INTEGER | Self-assessment score (e.g., 1-5 scale). |

confirms that the performance gains reported in Section 5 are not achieved through simple memorization of tool templates or domain-specific logic interpolation. Instead, the model must rely on the generalized reasoning and tool-invocation strategies internalized from the diverse ScaleEnv training environments to succeed in these semantically novel tasks.

## B. Detailed Statistics of Synthesized Domains and Tasks

### B.1. Visualization

We provide a comprehensive visualization of the 16 synthesized domains used in our training set. As illustrated in Figure 5, the domains exhibit significant structural diversity across three dimensions:

- **Action Space Scale (X-axis)**: The number of executable tools per domain ranges from approximately 25 (e.g., *Online Learning*) to over 70 (e.g., *Entertainment Media Query*), ensuring the agent learns to navigate varying sizes of action spaces.

- **State Space Complexity (Y-axis)**: The number of database tables ranges from 5 (e.g., *Job Seeking*) to 22 (e.g., *Agriculture Environment*), representing different levels of state-tracking difficulty.

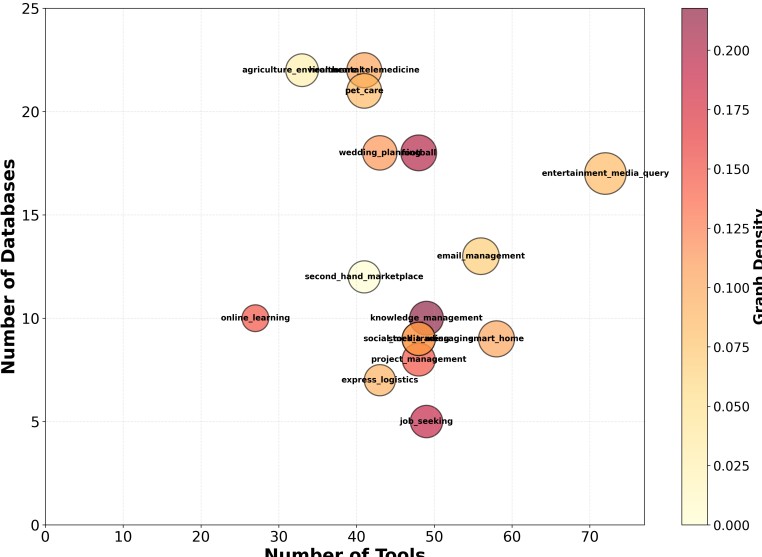

*Figure 5.* **Structural statistics of the 16 domains synthesized.** The x-axis and y-axis represent the number of tools and database tables, respectively. The color intensity and bubble size indicate the *Graph Density* of the Tool Dependency Graph, reflecting the complexity of inter-tool causal relationships within each domain.

- **Dependency Density (Color/Size)**: The color intensity represents the *Graph Density* of the Tool Dependency Graph. Domains like *Job Seeking* and *Knowledge Management* exhibit higher density (darker colors), indicating more complex inter-tool causal relationships, whereas others like *Agriculture Environment* are relatively sparser.

## C. Scalable RL in Hybrid Environments

### C.1. Method

To cultivate generalist agents, we construct a unified training universe $\mathbb{U} = \{(\mathcal{B}_k, \psi_j) \mid \psi_j \in \text{Tasks}(\mathcal{B}_k)\}$, where each instance pairs a specific synthesized domain environment $\mathbb{B}_k$ with a verifiable task $\psi_j$. We deploy an LLM-based user simulator, initialized with intent $\mathcal{U}$ from $\psi_j$, to generate natural language feedback $\mathcal{O}_{resp}$, closing the multi-turn interaction loop.

Within this high-fidelity POMDP, we optimize the agent policy $\pi_\theta$ using Group Relative Policy Optimization (GRPO) (Shao et al., 2024). For each task query $q \sim \mathbb{U}$, we sample a group of $G$ trajectories $\{o_1, \ldots, o_G\}$ from the old policy $\pi_{\theta_{old}}$. Recognizing the multi-step nature of tool use, we formulate the objective over time steps $t$ as:

$$\mathcal{J}(\theta) = \mathbb{E}_{q \sim \mathbb{U}} \left[ \frac{1}{G} \sum_{i=1}^{G} \frac{1}{L_i} \sum_{t=1}^{L_i} \left( \min\left( \rho_{i,t} \hat{A}_i, \text{clip}(\rho_{i,t}, 1-\epsilon, 1+\epsilon)\hat{A}_i \right) - \beta \mathbb{D}_{KL}(\pi_\theta(\cdot|h_{i,t})||\pi_{ref}(\cdot|h_{i,t})) \right) \right], \quad (1)$$

where $L_i$ is the trajectory length, and $\rho_{i,t} = \frac{\pi_\theta(a_{i,t}|h_{i,t})}{\pi_{\theta_{old}}(a_{i,t}|h_{i,t})}$ denotes the step-wise importance sampling weight given history $h_{i,t}$. Crucially, GRPO utilizes the group context to estimate the advantage $\hat{A}_i$. By normalizing the trajectory reward $r_i$ against the group statistics, we obtain a stable baseline without a value network as:

$$\hat{A}_i = \frac{r_i - \mu}{\sigma}, \quad (2)$$

where $\mu$ and $\sigma$ represent the mean and standard deviation of the intra-group rewards $\{r_1, \ldots, r_G\}$.

*Table 10.* Number of Tasks by Domain

| Domain Name | Number of Tasks |
| --- | --- |
| wedding_planning | 512 |
| knowledge_management | 512 |
| job_seeking | 256 |
| healthcare_telemedicine | 256 |
| pet_care | 128 |
| smart_home | 128 |
| email_management | 128 |
| football | 128 |
| second_hand_marketplace | 64 |
| express_logistics | 64 |
| social_media_messaging | 64 |
| entertainment_media_query | 64 |
| online_learning | 64 |
| stock_trading | 64 |
| project_management | 64 |
| agriculture_environmental | 64 |
| **Total** | 2560 |

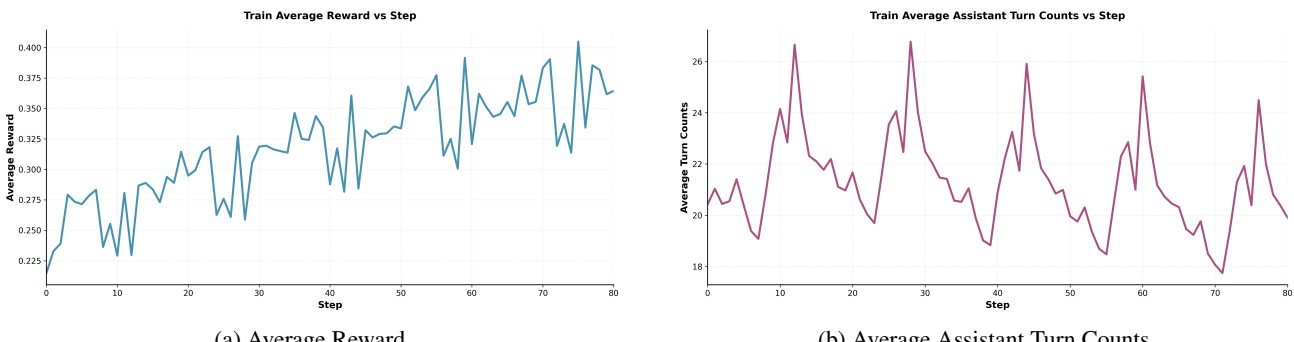

(a) Average Reward.  (b) Average Assistant Turn Counts.

*Figure 6.* **Training Dynamics of Qwen-3-8B on ScaleEnv.** (a) The average reward. (b) The assistant turn counts (excluding tool/user response).

## C.2. Training Dynamics Analysis

To analyze the stability and efficiency of our RL process, we visualize the training metrics of the **Qwen-3-8B** model over 5 epochs (80 optimization steps). The training utilized a dataset of 1024 synthesized tasks with a global rollout batch size of 1024 (calculated as $64 \times 16$ accumulation steps).

**Reward Convergence.** As illustrated in Figure 6(a), the average training reward exhibits a robust upward trajectory, climbing from approximately 0.22 to over 0.37. This steady convergence validates the efficacy of **ScaleEnv** as a high-quality training ground. By providing accurate, state-grounded supervision signals within a rich interactive environment, ScaleEnv enables the agent to progressively refine its policy and master complex tool-use strategies, directly translating to the observed performance improvements.

**Trajectory Efficiency.** Figure 6(b) tracks the average number of *assistant turns* per trajectory (excluding user queries and tool outputs). The turn count generally fluctuates around 20–26. Notably, we observe a slight downward trend in the later stages of training. We attribute this to the shift from exploration to exploitation: initially, the agent explores various tool combinations, resulting in longer interaction chains. As the policy matures, the agent learns to invoke tools with greater precision and correct parameters, thereby reducing redundant steps and solving tasks more efficiently.

# D. Additional Analysis

## D.1. Domain and Task Scaling

We systematically analyze the model performance trained from varying numbers of domains and tasks per domain. Table 11 reports the average performance across 4 domains in VitaBench.

*Table 11.* **Domain and task scaling analysis on VitaBench (Avg@4).** Performance improves with more domains and tasks, but exhibits diminishing returns at higher task counts.

|  | 16 Tasks/Dom. | 32 Tasks/Dom. | 64 Tasks/Dom. |
|---|---|---|---|
| 4 Domains | 10.2 | 12.5 | 12.1 |
| 8 Domains | 13.8 | 14.1 | 14.4 |
| 16 Domains | 13.9 | 14.6 | 14.5 |

We draw two main conclusions: (1) **Diminishing returns in task scaling.** Increasing tasks per domain from 16 to 32 yields noticeable improvements, but further scaling to 64 provides marginal gains, suggesting a saturation point in intra-domain task diversity. (2) **Domain expansion outweighs task expansion.** Expanding domains yields significantly higher returns than increasing tasks within existing domains. For instance, scaling tasks by $4\times$ ($16{\rightarrow}64$) at 4 domains only improves performance from 10.2 to 12.1, whereas scaling domains by $2\times$ ($4{\rightarrow}8$) at 16 tasks boosts performance to 13.8.

## D.2. Effect of the User Simulator

We investigate the impact of the user simulator's capacity by training Qwen3-8B with two different user simulators. As shown in Table 12, the overall impact of the user simulator's size on final performance is relatively limited. In practice, our primary criteria for selecting a user simulator are that it is sufficiently strong to follow instructions effectively while being deployable locally for large throughput.

*Table 12.* **Effect of user simulator on training.** Evaluated on 4 VitaBench domains.

| User Simulator | Train Reward | Avg Vita |
|---|---|---|
| Qwen3-32B | 0.21 | 13.7 |
| Qwen2.5-72B | 0.23 | 14.5 |

## D.3. Analysis of Dynamic Execution Paths

Table 13 reports trajectory statistics of Qwen3-32B on $\tau^2$-Bench, VitaBench, and 3 ScaleEnv domains: project_management (PM), football (FB), and express_logistics (EL).

*Table 13.* **Trajectory statistics across benchmarks and ScaleEnv domains.** Backtrack/branching rates denote the ratio of corresponding steps in a trajectory.

|  | PM | FB | EL | $\tau^2$-bench | VitaBench |
|---|---|---|---|---|---|
| Traj length (mean±std) | 35±17 | 25±17 | 25±21 | 30±20 | 47±25 |
| traj w/ backtrack (%) | 59 | 55 | 41 | 23 | 45 |
| Backtrack rate (%, succ/fail) | 4/3.2 | 2.4/6 | 2.5/2.6 | 1.3/1.3 | 1.3/1.9 |
| Branching rate (%) | 47 | 56 | 51 | 41 | 4 |

$\tau^2$-Bench produces trajectories with larger branching rates, while VitaBench has more trajectories with backtracking. Domains from ScaleEnv produce trajectories with high length variance, large branching rates, and frequent backtracking. Moreover, backtracking behavior has varying effects: in PM, successful trajectories backtrack more than failed ones, whereas in FB the relationship reverses. Such diverse patterns help training models generalize to different novel benchmarks.

## D.4. Generalization to DeepPlanning

To further validate that SCALEENV enables generalization to unseen domains without any possibility of data leakage, we evaluate on the recent DeepPlanning (Zhang et al., 2026) benchmark. As shown in Table 14, the ScaleEnv-trained model

substantially outperforms the base model, confirming robust generalization.

*Table 14.* **Generalization to DeepPlanning.**

| Model | Shopping (Match Score) | Travel (PS Score) |
|---|---|---|
| Qwen3-8B | 34.8 | 2.1 |
| Qwen3-8B + SCALEENV | 47.1 | 5.8 |

## D.5. Random Seed Stability

To verify that our results are stable across different random seeds, we train Qwen3-8B with 3 independent runs using different seeds. Table 15 reports the average performance on $\tau^2$-Bench and VitaBench.

*Table 15.* **Random seed stability of Qwen3-8B + ScaleEnv.** Results across 3 independent runs.

| Seed | $\tau^2$-Bench Avg | VitaBench Avg |
|---|---|---|
| Seed 1 | 38.5 | 15.0 |
| Seed 2 | 38.1 | 16.1 |
| Seed 3 | 37.7 | 15.2 |
| Mean±Std | 38.1±0.4 | 15.4±0.6 |

## E. Examples of Tool and Database

This section provides illustrative examples of the synthesized domain foundations for the "Job Seeking" domain. Table 9 presents the structured database schema, while Table 8 defines the functional tool interfaces. The corresponding executable implementations for both the database and the tools are provided in the subsequent listings.

```python
@with_instance_key("application_id")
class JobApplication(BaseModel, ThreadSafeBase["JobApplication"]):
    application_id: str = Field(..., description="Unique identifier for the job
    application")  # primary key
    applicant_name: str = Field(..., description="Full name of the job applicant")
    email: str = Field(..., description="Email address of the applicant")
    phone: Optional[str] = Field(default=None, description="Phone number of the applicant"
    )
    job_title: str = Field(..., description="Title of the job position being applied for")
    company_name: str = Field(..., description="Name of the company")
    application_date: datetime = Field(..., description="Date and time when the
    application was submitted")
    status: str = Field(default="submitted", description="Current status of the
    application (e.g., submitted, under_review, rejected, accepted, withdrawn, archived)")
    resume_content: Optional[str] = Field(default=None, description="Text content of the
    resume")
    resume_format: Optional[str] = Field(default=None, description="Format of the resume
    document (e.g., pdf, docx)")
    resume_uploaded_at: Optional[datetime] = Field(default=None, description="Timestamp
    when the resume was uploaded")
    cover_letter_content: Optional[str] = Field(default=None, description="Text content of
     the cover letter")
    cover_letter_uploaded_at: Optional[datetime] = Field(default=None, description="
    Timestamp when the cover letter was uploaded")
    deadline_date: Optional[datetime] = Field(default=None, description="Follow-up or
    response deadline for the application")
    deadline_type: Optional[str] = Field(default=None, description="Type of deadline (e.g
    ., follow_up, response)")
    referral_source: Optional[str] = Field(default=None, description="Source of the job
    referral (e.g., LinkedIn, Indeed, employee referral)")
    referral_person: Optional[str] = Field(default=None, description="Name of person who
    referred the applicant")
```

```python
    priority_level: Optional[int] = Field(default=None, description="Priority level of the
     application (1-5, where 5 is highest)")
    priority_reason: Optional[str] = Field(default=None, description="Reason for the
    assigned priority level")
    expected_salary_min: Optional[float] = Field(default=None, description="Minimum
    expected salary")
    expected_salary_max: Optional[float] = Field(default=None, description="Maximum
    expected salary")
    salary_currency: Optional[str] = Field(default="USD", description="Currency for salary
     expectations (e.g., USD, EUR)")
    created_at: datetime = Field(..., description="Timestamp when the record was created")
    updated_at: Optional[datetime] = Field(default=None, description="Timestamp when the
    record was last updated")

@with_instance_key("note_id")
class ApplicationNote(BaseModel, ThreadSafeBase["ApplicationNote"]):
    note_id: str = Field(default=..., description="Unique identifier for the note")
    application_id: str = Field(default=..., description="Reference to the associated job
    application")
    note_content: str = Field(default=..., description="Content of the note or comment")
    note_type: Optional[str] = Field(default=None, description="Type or category of the
    note (e.g., follow_up, reminder, general)")
    created_at: datetime = Field(default=..., description="Timestamp when the note was
    created")

@with_instance_key("stage_id")
class ApplicationStage(BaseModel, ThreadSafeBase["ApplicationStage"]):
    stage_id: str = Field(default=..., description="Unique identifier for the stage record
    ")
    application_id: str = Field(default=..., description="Reference to the associated job
    application")
    stage_name: str = Field(default=..., description="Name of the application stage (e.g.,
     phone_screening, technical_interview, final_interview)")
    stage_date: datetime = Field(default=..., description="Date and time when the stage
    was reached")
    stage_notes: Optional[str] = Field(default=None, description="Additional notes about
    the stage")  # stage_notes is optional

@with_instance_key("interview_id")
class InterviewSchedule(BaseModel, ThreadSafeBase["InterviewSchedule"]):
    interview_id: str = Field(default=..., description="Unique identifier for the
    interview schedule")
    application_id: str = Field(default=..., description="Reference to the associated job
    application")
    interview_type: str = Field(default=..., description="Type of interview (e.g.,
    phone_screening, technical_interview, behavioral_interview)")
    interview_date: datetime = Field(default=..., description="Scheduled date and time of
    the interview")
    interviewer_name: Optional[str] = Field(default=None, description="Name of the
    interviewer")
    interview_location: Optional[str] = Field(default=None, description="Location or
    platform for the interview (e.g., Zoom, office address)")
    interview_duration_minutes: Optional[int] = Field(default=None, description="Expected
    duration of the interview in minutes")

@with_instance_key("feedback_id")
class InterviewFeedback(BaseModel, ThreadSafeBase["InterviewFeedback"]):
    feedback_id: str = Field(default=..., description="Unique identifier for the interview
     feedback")
    interview_id: str = Field(default=..., description="Reference to the associated
    interview")
    feedback_content: str = Field(default=..., description="Content of the feedback or
    notes")
    performance_rating: Optional[int] = Field(default=None, description="Self-assessment
    rating of interview performance (e.g., 1-5 scale)")
```

```python
    created_at: datetime = Field(default=..., description="Timestamp when the feedback was
     created")

class JobSeekingDB(DB):
    """Database containing all job_seeking_Job_Application-related data"""
    job_application: Optional[Dict[str, JobApplication]] = Field(
        default=None,
        description="Schema JobApplication"
    )
    application_note: Optional[Dict[str, ApplicationNote]] = Field(
        default=None,
        description="Schema ApplicationNote"
    )
    application_stage: Optional[Dict[str, ApplicationStage]] = Field(
        default=None,
        description="Schema ApplicationStage"
    )
    interview_schedule: Optional[Dict[str, InterviewSchedule]] = Field(
        default=None,
        description="Schema InterviewSchedule"
    )
    interview_feedback: Optional[Dict[str, InterviewFeedback]] = Field(
        default=None,
        description="Schema InterviewFeedback"
    )
```

*Listing 1.* Python implementation for database code in job seeking domain.

```python
# ==================== 1. delete_job_application ====================
def delete_job_application(self, application_id: str) -> dict:
    """
    Delete a job application from the system

    Args:
        application_id: Unique identifier of the application to delete

    Returns:
        Dictionary containing:
        - application_id: Unique identifier of the deleted application
        - deletion_status: Status of the deletion operation
        - deleted_at: Timestamp when the application was deleted in yyyy-mm-dd HH:MM:SS
    format

    Raises:
        KeyError: If the application_id does not exist in the system
    """
    # Validate input parameter
    if not application_id or not isinstance(application_id, str):
        raise ValueError("application_id must be a non-empty string")

    # Access the database
    db = self.db

    # Get the job_application table
    job_application_table = getattr(db, 'job_application', None)

    # Check if the table exists
    if job_application_table is None:
        raise KeyError(f"Application with ID '{application_id}' does not exist in the
    system")

    # Check if the application exists in the table
    if application_id not in job_application_table:
        raise KeyError(f"Application with ID '{application_id}' does not exist in the
    system")
```

```python
    # Remove the application from the table
    updated_table = {k: v for k, v in job_application_table.items() if k != application_id
    }

    # Update the database table with the new dictionary
    setattr(db, 'job_application', updated_table)

    # Generate deletion timestamp
    deleted_at = datetime.now().strftime("%Y-%m-%d %H:%M:%S")

    # Return the deletion result
    return {
        'application_id': application_id,
        'deletion_status': 'deleted',
        'deleted_at': deleted_at
    }

# ==================== 2. batch_update_application_status ====================
def batch_update_application_status(self, application_ids: list, new_status: str,
    updated_at: str) -> dict:
    """
    Update status for multiple applications at once

    Args:
        application_ids: List of application identifiers to update
        new_status: New status to apply to all applications
        updated_at: Timestamp of the batch update in yyyy-mm-dd HH:MM:SS format

    Returns:
        dict: Contains updated_count (number of successfully updated applications)
              and failed_updates (list of application IDs that failed to update)

    Raises:
        ValueError: If parameters are invalid or applications don't exist
    """
    # Validate input parameters
    if not application_ids:
        raise ValueError("application_ids cannot be empty")

    if not isinstance(application_ids, list):
        raise ValueError("application_ids must be a list")

    if not new_status or not isinstance(new_status, str):
        raise ValueError("new_status must be a non-empty string")

    if not updated_at or not isinstance(updated_at, str):
        raise ValueError("updated_at must be a non-empty string")

    # Parse and validate the updated_at timestamp
    try:
        update_datetime = datetime.strptime(updated_at, "%Y-%m-%d %H:%M:%S")
        formatted_updated_at = update_datetime.strftime("%Y-%m-%d %H:%M:%S")
    except ValueError:
        raise ValueError("updated_at must be in 'yyyy-mm-dd HH:MM:SS' format")

    # Access the database
    db = self.db
    job_application_table = getattr(db, "job_application", None)

    if job_application_table is None:
        raise ValueError("job_application table not found in database")

    # Initialize tracking variables
```

```python
    updated_count = 0
    failed_updates = []

    # Process each application ID
    for app_id in application_ids:
        try:
            # Check if the application exists
            if app_id not in job_application_table:
                # Application doesn't exist, add to failed list
                failed_updates.append(app_id)
                continue

            # Retrieve the application record
            application = job_application_table[app_id]

            # Update the status and updated_at timestamp (as string)
            application.status = new_status
            application.updated_at = formatted_updated_at

            # Save the updated application back to the database
            job_application_table[app_id] = application

            # Increment successful update count
            updated_count += 1

        except Exception:
            # If any error occurs during update, add to failed list
            failed_updates.append(app_id)
            continue

    # Update the database table
    setattr(db, "job_application", job_application_table)

    # Return the results
    return {
        "updated_count": updated_count,
        "failed_updates": failed_updates
    }

# ===================== 3. archive_old_applications =====================
def archive_old_applications(self, cutoff_date: str, archive_status: str = "archived") ->
    dict:
    """
    Archive applications older than a specified date by updating their status.

    Args:
        cutoff_date: Date before which applications should be archived in yyyy-mm-dd
    format
        archive_status: Status to set for archived applications (default: 'archived')

    Returns:
        dict: Contains 'archived_count' (number of applications archived) and
              'archived_application_ids' (list of archived application IDs)

    Raises:
        ValueError: If cutoff_date format is invalid or if no job_application table exists
    """
    # Validate cutoff_date format
    try:
        cutoff_datetime = datetime.strptime(cutoff_date, "%Y-%m-%d")
    except ValueError as e:
        raise ValueError(f"Invalid cutoff_date format. Expected yyyy-mm-dd, got: {
    cutoff_date}") from e
```

```python
    # Access the database
    db = self.db

    # Get the job_application table
    job_application_table = getattr(db, "job_application", None)

    # Validate that the table exists
    if job_application_table is None:
        raise ValueError("job_application table does not exist in the database")

    # Initialize tracking variables
    archived_count = 0
    archived_application_ids = []

    # Iterate through all applications in the table
    for application_id, application in job_application_table.items():
        # Check if application_date is before the cutoff_date
        if application.application_date.date() < cutoff_datetime.date():
            # Update the application status to archive_status
            application.status = archive_status

            # Update the updated_at timestamp to current time
            application.updated_at = datetime.now()

            # Track the archived application
            archived_count += 1
            archived_application_ids.append(application_id)

    # Write the updated table back to the database
    setattr(db, "job_application", job_application_table)

    # Return the results
    return {
        "archived_count": archived_count,
        "archived_application_ids": archived_application_ids
    }
```

*Listing 2.* Part of Python implementation for tool code in job seeking domain.

## F. Agent-User Interaction Trajectory in ScaleEnv

**Assistant [1]**

Hello, how can I help you?

**User [2]**

Hi, I've been applying to several energy analyst positions recently, and it's getting a bit chaotic. Could you help me check the status of all the energy analyst-related positions I've applied for? I want to make sure I have a clear overview.

**Assistant [3] Tool Calls:**

**Tool 1: search_applications_by_keyword**
Arguments: {"keyword": "energy analyst", "search_fields": ["job_title"]}

**Tool/Response [4]**

{"matching_applications": [{"application_id": "APP001", "job_title": "Senior Energy Analyst", "company_name": "PetroChina Group"}, {"application_id": "APP002", "job_title": "Energy Market Analyst", "company_name": "CNOOC"}, {"application_id": "APP003", "job_title": "New Energy Strategic Analyst", "company_name": "Envision Energy"}, {"application_id": "APP004", "job_title": "Data Analyst", "company_name": "Tencent Technology"}, {"application_id": "APP005", "job_title": "Energy Investment Consulta...

**Assistant [5] Tool Calls:**

**Tool 1: get_application_stage_history**
Arguments: {"application_id": "APP001"}
**Tool 2: get_application_stage_history**
Arguments: {"application_id": "APP002"}
**Tool 3: get_application_stage_history**
Arguments: {"application_id": "APP003"}
**Tool 4: get_application_stage_history**
Arguments: {"application_id": "APP004"}
**Tool 5: get_application_stage_history**
Arguments: {"application_id": "APP005"}
**Tool 6: get_application_stage_history**
Arguments: {"application_id": "APP007"}
**Tool 7: get_application_stage_history**
Arguments: {"application_id": "APP008"}
**Tool 8: get_application_stage_history**
Arguments: {"application_id": "APP009"}

**Tool/Response [6]**

{"stages": [{"stage_id": "STAGE001", "stage_name": "Resume screening passed", "stage_date": "2024-02-25 15:00:00", "stage_notes": "HR notified that the resume passed the initial screening, waiting for interview arrangement"}, {"stage_id": "STAGE002", "stage_name": "Waiting for technical interview", "stage_date": "2024-03-10 10:00:00", "stage_notes": "Received interview invitation email, confirmed interview time"}]}

**Tool/Response [7]**

{"stages": [{"stage_id": "STAGE003", "stage_name": "Resume screening passed", "stage_date": "2024-03-01 09:00:00", "stage_notes": "Initial HR phone communication"}, {"stage_id": "STAGE004", "stage_name": "Phone Interview Completed", "stage_date": "2024-03-08 10:30:00", "stage_notes": "Performed well in the phone interview, awaiting next round notification"}]}

**Tool/Response [8]**

{"stages": [{"stage_id": "STAGE005", "stage_name": "Resume Submitted", "stage_date": "2024-03-01 09:20:00", "stage_notes": "Resume submitted via official website"}]}

**Tool/Response [9]**

{"stages": [{"stage_id": "STAGE006", "stage_name": "Resume Screening Passed", "stage_date": "2024-02-20 14:00:00", "stage_notes": "Received interview invitation"}, {"stage_id": "STAGE007", "stage_name": "Technical Interview Completed", "stage_date": "2024-02-25 11:00:00", "stage_notes": "Average interview performance"}, {"stage_id": "STAGE008", "stage_name": "Application Rejected", "stage_date": "2024-02-28 10:30:00", "stage_notes": "Received rejection letter, feedback was insufficient industry ...

**Tool/Response [10]**

{"stages": [{"stage_id": "STAGE009", "stage_name": "Resume Screening Passed", "stage_date": "2024-03-05 11:00:00", "stage_notes": "Referral assisted with internal recommendation, HR responded quickly"}, {"stage_id": "STAGE010", "stage_name": "Phone Interview Completed", "stage_date": "2024-03-16 14:45:00", "stage_notes": "Initial communication was effective"}]}

**Tool/Response [11]**

{"stages": [{"stage_id": "STAGE013", "stage_name": "Resume Submitted", "stage_date": "2024-03-08 14:00:00", "stage_notes": "Submitted via job portal"}]}

**Tool/Response [12]**

{"stages": [{"stage_id": "STAGE014", "stage_name": "Resume Submitted", "stage_date": "2024-03-10 16:30:00", "stage_notes": "Submitted through alumni referral"}]}

**Tool/Response [13]**

{"stages": [{"stage_id": "STAGE015", "stage_name": "Resume Submitted", "stage_date": "2024-03-12 09:00:00", "stage_notes": "Submitted via government recruitment website"}, {"stage_id": "STAGE016", "stage_name": "Resume Screening Passed", "stage_date": "2024-03-18 15:00:00", "stage_notes": "Received interview notification"}]}

**Assistant [14]**

Here's a clear overview of all your energy analyst-related applications:
— # — Position — Company — Current Status — Last Update — ————————————————————————————————————— 1 — **Senior Energy Analyst** — PetroChina Group — Waiting for Technical Interview — 2024-03-10 — — 2 — **Energy Market Analyst** — CNOOC — Phone Interview Completed (awaiting next round) — 2024-03-08 — — 3 — **New Energy Strategic Analyst** — Envision Energy — Resume Submitted (pending review) — 2024-03-01 — — 4 — **Data Analyst** — Tencent Technology — Rejected — 2024-02-28 — — 5 — **Energy Investment Consultant** — CITIC Securities — Phone Interview Completed — 2024-03-16 — — 6 — **Strategic Analysis Manager** — State Grid Corporation — Resume Submitted (pending review) — 2024-03-08 — —...

**User [15]**

Thanks for the summary! I'm particularly interested in the Senior Energy Analyst position at PetroChina. Their HR mentioned they can arrange a technical interview for me next Monday at 2 PM, at Zhongguancun Software Park in Haidian, with Manager Zhang as the interviewer, and the interview will last about an hour. Could you help me record this information and set a reminder for the interview?

**Assistant [16] Tool Calls:**

**Tool 1: add_interview_schedule**
Arguments: {"application_id": "APP001", "interview_date": "2024-03-18 14:00:00", "interview_duration_minutes": 60, "interview_location": "Zhongguancun Software Park, Haidian", "interview_type": "technical_interv

**Tool 2: add_application_note**
Arguments: {"application_id": "APP001", "created_at": "2024-03-15 09:30:00", "note_content": "INTERVIEW REMINDER: Technical interview scheduled for Monday, March 18, 2024 at 2:00 PM. Location: Zhongguancun Softw

**Tool 3: calculate_interview_preparation_time**
Arguments: {"current_date": "2024-03-15 09:30:00", "interview_date": "2024-03-18 14:00:00"}

**Tool/Response [17]**

{"interview_id": "INT-0550001", "application_id": "APP001"}

**Tool/Response [18]**

{"note_id": "NOTE-2024-235720", "application_id": "APP001"}

**Tool/Response [19]**

{"days_until_interview": "3", "hours_until_interview": "76", "is_urgent": "False"}

**Assistant [20]**

I've recorded all the interview information and set a reminder for you. Here's a summary:
### Interview Scheduled Successfully
— Detail — Information — ————————————— — **Position** — Senior Energy Analyst — — **Company** — PetroChina Group — — **Date & Time** — Monday, March 18, 2024 at 2:00 PM — — **Location** — Zhongguancun Software Park, Haidian — — **Interviewer** — Manager Zhang — — **Duration** — ~60 minutes — — **Interview Type** — Technical Interview — — **Interview ID** — INT-0550001 —
### Preparation Time You have **3 days (about 76 hours)** to prepare for this interview - plenty of time to get ready!
### Reminder Note Added A reminder note has been added to your application with all the key details.
**Suggested Preparation Tips:** - Review PetroChina's rece...

**User [21]**

Thanks for setting up the reminder! I also want to record my impressions of the interview I finished last week. The overall conversation was decent, and we had an in-depth discussion on industry trends, but I didn't answer the specific model details very well and need to improve. I'd give myself a score of 4 for that interview. Could you help me document this feedback?

**Assistant [22] Tool Calls:**

**Tool 1: get_application_interviews**
Arguments: {"application_id": "APP002"}

**Tool/Response [23]**

{"interviews": [{"interview_id": "INT002", "interview_type": "phone_screening", "interview_date": "2024-03-08 10:00:00", "interviewer_name": "Liu HR", "interview_location": "Phone interview"}]}

**Assistant [24] Tool Calls:**

**Tool 1: add_interview_feedback**
Arguments: {"created_at": "2024-03-15 09:30:00", "feedback_content": "Overall the conversation went well. Had an in-depth discussion on industry trends which was productive. Area for improvement: Did not answer

**Tool/Response [25]**

{"feedback_id": "FEEDBACK-2024-0EDA01", "interview_id": "INT002"}

**Assistant [26]**

I've documented your interview feedback. Here's a summary:
### Interview Feedback Recorded
— Detail — Information — —————————–— — **Position** — Energy Market Analyst — — **Company** — — CNOOC — — **Interview Date** — March 8, 2024 (Phone Screening) — — **Self-Rating** — 4/5 — **Feedback Summary:** — Strengths — Areas for Improvement — —————————–— — Overall conversation went well — Specific model details need work — — In-depth discussion on industry trends — Need to strengthen technical modeling knowledge —
This feedback will be helpful for your upcoming PetroChina technical interview on Monday! You might want to spend some of your preparation time reviewing specific analytical models and technical details.
Is there anything else you'd ...

**User [27]**

Thanks for recording the feedback. I'd like to analyze my job search efficiency during this period. Could you help me calculate my interview conversion rate and offer conversion rate? I want to see if I need to adjust my strategy based on these metrics.

**Assistant [28] Tool Calls:**

**Tool 1: calculate_application_roi**
Arguments: {"total_applications": 8, "total_interviews": 5, "total_offers": 0}
**Tool 2: calculate_application_success_rate**
Arguments: {"rejected_applications": 1, "successful_applications": 0, "total_applications": 8}

**Tool/Response [29]**

{"interview_conversion_rate": "62.5", "offer_conversion_rate": "0.0", "interview_to_offer_rate": "0.0"}

**Tool/Response [30]**

{"success_rate": "0.0", "rejection_rate": "12.5", "pending_rate": "87.5"}

**Assistant [31]**

Here's a comprehensive analysis of your job search efficiency:
### Job Search Performance Metrics
#### Conversion Rates — Metric — Rate — Interpretation — ————————————————–— — **Interview Conversion Rate** — **62.5%** — 5 out of 8 applications reached interview stage — — **Offer Conversion Rate** — 0% — No offers received yet — — **Interview → Offer Rate** — 0% — Still in progress —
#### Application Status Breakdown — Status — Count — Percentage — ———————————-————— — Pending/In Progress — 7 — 87.5% — — Rejected — 1 — 12.5% — — Offers — 0 — 0% — —

### Analysis & Insights
**Strengths:** - **Strong resume performance** - Your 62.5% interview conversion rate is excellent! This means your resume and qualifications are well-aligned with energy an...

**User [32]**

Thanks for the detailed analysis! Given my current interview conversion rate and the feedback I received, I think I should focus on improving my technical modeling skills. Could you recommend any specific resources or practice methods that would be most effective for preparing for technical interviews in the energy industry?

**Assistant [33]**

I appreciate you asking, but recommending specific learning resources or study methods is outside the scope of what I can help with in this job application tracking system. My tools are focused on managing and tracking your job applications, interviews, and related activities.
However, there are a few things I **can** help you with to support your preparation:
1. **Add a preparation note** to your PetroChina application with specific technical topics you want to review before Monday's interview
2. **Set a follow-up reminder** to track your study progress
3. **Add action items** to your applications noting areas to improve based on interview feedback
4. **Review your application materials** - I could help you check if your resume highlights relevant technical/modeling skills
Would any...

**User [34]**

Thanks for the suggestions! I'd like to add a preparation note to my PetroChina application, focusing on specific technical topics like energy market models, data analysis tools, and recent industry trends. Could you help me draft a concise note for this?

**Assistant [35] Tool Calls:**

**Tool 1: add_application_note**
Arguments: {"application_id": "APP001", "created_at": "2024-03-15 09:30:00", "note_content": "TECHNICAL INTERVIEW PREPARATION CHECKLIST:\{}n\{}n1. ENERGY MARKET MODELS:\{}n - Review supply/demand forecasting models\{}

**Tool/Response [36]**

{"note_id": "NOTE-2024-AF1DEF", "application_id": "APP001"}

### Assistant [37]

I've added a detailed preparation note to your PetroChina application. Here's what I documented:
### Preparation Note Added
**Application:** Senior Energy Analyst @ PetroChina Group **Note Type:** Action Item **Interview Date:** Monday, March 18, 2024 at 2:00 PM
—

### Technical Interview Preparation Checklist
— Topic — Key Focus Areas — ——————————————— — **1. Energy Market Models** — • Supply/demand forecasting models¡br¿• Price elasticity & market equilibrium¡br¿• Examples of models you've worked with — — **2. Data Analysis Tools** — • Excel (advanced functions, pivot tables)¡br¿• Python/R for data analysis¡br¿• Energy-specific software/databases — — **3. Recent Industry Trends** — • PetroChina's latest projects & strategy¡br¿• Current oil & gas market conditions¡br...

### User [38]

Thanks for the preparation note! I'd also like to set a follow-up reminder for my other pending applications, especially for Envision Energy, State Grid Corporation, and Procter & Gamble. Could you help me schedule these reminders?

### Assistant [39] Tool Calls:

**Tool 1: generate_follow_up_reminder**
Arguments: {"application_date": "2024-03-01 09:20:00", "company_name": "Envision Energy", "follow_up_date": "2024-03-18 10:00:00", "job_title": "New Energy Strategic Analyst"}
**Tool 2: generate_follow_up_reminder**
Arguments: {"application_date": "2024-03-08 14:00:00", "company_name": "State Grid Corporation of China", "follow_up_date": "2024-03-20 10:00:00", "job_title": "Strategic Analysis Manager"}
**Tool 3: generate_follow_up_reminder**
Arguments: {"application_date": "2024-03-10 16:30:00", "company_name": "Procter & Gamble", "follow_up_date": "2024-03-22 10:00:00", "job_title": "Market Research Analyst"}

### Tool/Response [40]

{"reminder_message": "Follow up on your New Energy Strategic Analyst application at Envision Energy (applied on 2024-03-01)"}

### Tool/Response [41]

{"reminder_message": "Follow up on your Strategic Analysis Manager application at State Grid Corporation of China (applied on 2024-03-08)"}

### Tool/Response [42]

{"reminder_message": "Follow up on your Market Research Analyst application at Procter & Gamble (applied on 2024-03-10)"}

**Assistant [43] Tool Calls:**

**Tool 1: add_application_note**
Arguments: {"application_id": "APP003", "created_at": "2024-03-15 09:30:00", "note_content": "FOLLOW-UP REMINDER: Contact Envision Energy on Monday, March 18, 2024 to check application status. It will be 17 days

**Tool 2: add_application_note**
Arguments: {"application_id": "APP007", "created_at": "2024-03-15 09:30:00", "note_content": "FOLLOW-UP REMINDER: Contact State Grid Corporation on Wednesday, March 20, 2024 to check application status. It will

**Tool 3: add_application_note**
Arguments: {"application_id": "APP008", "created_at": "2024-03-15 09:30:00", "note_content": "FOLLOW-UP REMINDER: Contact Procter & Gamble on Friday, March 22, 2024 to check application status. It will be 12 day

**Tool 4: set_application_deadline**
Arguments: {"application_id": "APP003", "deadline_date": "2024-03-18 10:00:00", "deadline_type": "follow_up"}

**Tool 5: set_application_deadline**
Arguments: {"application_id": "APP007", "deadline_date": "2024-03-20 10:00:00", "deadline_type": "follow_up"}

**Tool 6: set_application_deadline**
Arguments: {"application_id": "APP008", "deadline_date": "2024-03-22 10:00:00", "deadline_type": "follow_up"}

**Tool/Response [44]**

{"note_id": "NOTE-2024-369CF6", "application_id": "APP003"}

**Tool/Response [45]**

{"note_id": "NOTE-2024-45E718", "application_id": "APP007"}

**Tool/Response [46]**

{"note_id": "NOTE-2024-B579B8", "application_id": "APP008"}

**Tool/Response [47]**

{"application_id": "APP003", "deadline_set": "True"}

**Tool/Response [48]**

{"application_id": "APP007", "deadline_set": "True"}

**Tool/Response [49]**

{"application_id": "APP008", "deadline_set": "True"}

**Assistant [50]**

I've set up follow-up reminders and deadlines for all three pending applications. Here's your follow-up schedule:
### Follow-Up Reminders Scheduled
— # — Company — Position — Applied On — Follow-Up Date — Days Since Applied — ————————————————— ————————————————————————————————————— — 1 — **Envision Energy** — New Energy Strategic Analyst — Mar 1 — **Mon, Mar 18** — 17 days — — 2 — **State Grid Corporation** — Strategic Analysis Manager — Mar 8 — **Wed, Mar 20** — 12 days — — 3 — **Procter & Gamble** — Market Research Analyst — Mar 10 — **Fri, Mar 22** — 12 days —
—
### Your Week Ahead (Mar 15-22)
— Date — Action Items — ————————————— — **Mon, Mar 18** — Technical Interview @ PetroChina (2:00 PM) Follow up with Envision Energy — — **We...

**User [51]**

Thanks for setting up the follow-up reminders! I think that covers everything I needed for now. If I have any more updates or need further assistance, I'll reach out.

**Assistant [52]**

You're welcome! Here's a quick summary of everything we accomplished today:
### Session Summary
— Task — Status — ———————— — Reviewed all 8 energy analyst applications — Complete — — Scheduled PetroChina technical interview (Mar 18, 2:00 PM) — Complete — — Added interview reminder for PetroChina — Complete — — Recorded CNOOC interview feedback (Rating: 4/5) — Complete — — Analyzed job search metrics (62.5% interview conversion rate) — Complete — — Added technical preparation checklist for PetroChina — Complete — — Set follow-up reminders for 3 pending applications — Complete —
### Upcoming Reminders - **Mon, Mar 18** - PetroChina Interview + Envision Energy follow-up - **Wed, Mar 20** - State Grid Corporation follow-up - **Fri, Mar 22** - Procter & Gamble foll...

