# OpenReview forum: "ScaleEnv: Scaling Environment Synthesis from Scratch for Generalist Interactive Tool-Use Agent Training"
_ICML.cc/2026/Conference — ICML 2026 regular_

### Official Review · Reviewer_84SD · 2026-03-11

**Soundness:** 2
**Presentation:** 2
**Significance:** 3
**Originality:** 3
**Overall Recommendation:** 3
**Confidence:** 5

**Summary:**

This paper studies how to train generalist tool-use agents with reinforcement learning when high-quality interactive environments are scarce. Existing approaches often rely on LLM-based simulators, which may suffer from hallucinations, limited controllability, and poor scalability.

To address this, the authors propose ScaleEnv, a framework that programmatically generates interactive and verifiable environments using Python tools and databases. The method first constructs tool implementations and database schemas via multi-agent generation with procedural testing, forming a tool dependency graph. It then applies graph expansion and execution-based verification to synthesize database states that ensure task solvability while supporting exploratory interactions. The framework also uses rule-based rewards derived from database state comparisons rather than LLM judges.

Agents trained in ScaleEnv with GRPO show improved zero-shot generalization on unseen benchmarks such as τ²-Bench and VitaBench, and experiments suggest that increasing environment diversity improves performance more effectively than simply scaling task counts.

**Compliance With Llm Reviewing Policy:**

Affirmed.

**Final Justification:**

I think the extensive experiments will be useful for the rest of the community, while I might not be 100% about its innovation.

**Key Questions For Authors:**

1. The paper claims in the introduction that EnvScaler is limited to single-turn user–agent interaction. This appears to be inaccurate. The EnvScaler paper explicitly emphasises improvements in multi-turn, multi-tool interactions (see both the abstract and main text). The authors should carefully revisit the description of EnvScaler and ensure that related work is represented accurately.

2. The ablation in Table 4 comparing rule-based rewards with LLM-as-a-judge is interesting. However, it is unclear whether the comparison implicitly favors rule-based evaluation because the generated environments have structured, easily verifiable database states. The authors could clarify the scope and limitations of the rule-based reward design—for example, whether it would still apply to tasks with less structured or harder-to-formalize success criteria. A brief discussion would help address potential fairness concerns in this comparison.

3. Section 4.2.2 relies on Qwen3-235B with Best-of-16 sampling as an oracle for feasibility checking. This raises a question about the cost-efficiency of the proposed scaling approach: if environment generation requires repeated sampling from a 200B+ model, the overall computational cost may be substantial. It would strengthen the paper if the authors provided a rough estimate of the generation cost (e.g., average tokens, runtime, or GPU hours per domain/task). Including such statistics in the appendix would help readers assess the practical scalability of the framework.

4. The graph expansion strategy may lead to potential **state explosion** as environments scale. While the paper enforces constraints such as ( $|\mathcal{H}_n| \ge 20$ ), expanding tool logic and database schemas could introduce inconsistencies or synthesis failures. The paper does not report the environment synthesis failure rate. It would improve transparency to report whether any domains failed during generation (e.g., among the 16 domains used) and the rate of discarded or invalid domains, which would clarify the practical limits of the pipeline.

5. The diversity analysis in Figure 4 relies on t-SNE over tool descriptions, which mainly reflects semantic differences. It is unclear whether the environments also differ in underlying graph topology or reasoning structure, rather than simply varying domain surface forms (e.g., flight vs. telecom) while sharing similar interaction patterns. It would strengthen the paper to analyze structural diversity, such as differences in tool-dependency graphs or interaction trajectories, to confirm that the generated domains are not merely template variations.

**Limitations:**

The paper includes an Ethics section but does not explicitly discuss **technical limitations**. For example, the proposed pipeline relies on a very large **oracle model (Qwen3-235B)** during environment generation, which raises potential concerns about **computational cost and accessibility**. In addition, the framework assumes environments that can be represented through **discrete database states and tool executions**. This design may limit applicability to domains involving **continuous dynamics or complex physical control**, where outcomes cannot be easily verified through simple database state comparisons. A dedicated discussion of these limitations would help clarify the **practical scope and boundaries** of the proposed approach.

**Strengths And Weaknesses:**

**Soundness**
- Strengths
1. The paper includes well-designed ablations for **Execution Verification (EV)** and the **reward mechanism** (Tables 3 and 4), which provide useful evidence that these components contribute to performance improvements.
2. The evaluation clearly investigates the relationship between **rule-based evaluation** and generalization performance.
3. The EV and reward ablations are carefully targeted and directly test the key mechanisms proposed in the paper.

- Weaknesses
1. While the paper analyzes **semantic diversity** across domains, it is unclear whether the **underlying interaction patterns** differ from those in the evaluation benchmarks (τ²-Bench and VitaBench). The current analysis focuses on semantic distance rather than structural differences in interaction graphs or reasoning patterns.
2. The paper lacks **direct comparisons with other synthetic environment or dataset construction approaches** (e.g., EnvScaler, Simia) or strong SFT-based baselines such as ToolACE using the same base model. Without these comparisons, the evaluation may appear to rely on relatively weak baselines.
3. The paper does not report the **failure rate of environment synthesis**. During graph expansion, domains could potentially fail due to logical inconsistencies, deadlocks, or database conflicts. Reporting how often generation fails (or domains are discarded) would clarify the robustness of the pipeline.
4. RL training is reported for **48 steps**, but the paper does not provide a **training curve** showing reward or performance versus training steps. Without this, it is unclear whether the model has converged or whether results may depend on a particular stopping point.
5. Table 1 compares against several open models, but for **Qwen3**, the comparison is only against the base model. Stronger baselines such as **Qwen3 + SFT (ToolACE)** or **Qwen3 + Simia-style data** would help isolate whether the gains come from the proposed environment or simply from applying RL.
6. None of the RL experiments (Tables 1, 3, 4, 5) report **variance across random seeds**. Given the instability of RL training, reporting multiple runs or confidence intervals would significantly strengthen the reliability of the results.
7. Although the paper studies **domain scaling** (Table 5), it does not analyze the sensitivity of key generation parameters (e.g., **distractor injection density**) on downstream RL performance.


**Presentation**
- Strengths
1. **Figures 1 and 2** are particularly clear and effectively illustrate the system pipeline and environment generation process.

- Weaknesses
1. The paper does not report the **token cost or wall-clock time** required for environment synthesis, which is important for assessing the practical scalability of the approach.
2. The selection process for the **16 training domains** is not fully specified; a clearer description or formal selection criterion would improve reproducibility.
3. The synthesis phase mentions a **diverse suite of high-performance LLMs** serving different agent roles, but the coordination strategy and role assignment among these models are not described in sufficient detail.


**Significance**

The paper addresses a **core bottleneck in the agent research ecosystem**: the scarcity of scalable, high-quality, and interactive environments that support reinforcement learning and large-scale exploration. If validated, the proposed framework could provide a useful infrastructure for training tool-use agents.

**Originality**
- Strengths
1. Introducing **interaction completeness** as an explicit optimization objective for environment generation is a compelling idea.
2. The use of **graph expansion** to enforce solvability while preserving exploratory interaction is a novel perspective.

- Weaknesses
1. The observation that **LLM-as-a-judge can lead to reward hacking**, motivating a return to rule-based verification, appears to be an insight derived from empirical failure rather than a fundamentally new conceptual contribution.

---

> ### Author Rebuttal · Authors · 2026-03-31
>
> We sincerely thank the reviewer for recognizing the clarity of our presentation and the significance to address environment scarcity for agentic RL. Below we address your thoughtful questions and concerns point by point.
> ### Q1 (EnvScaler)
> Our original description in introduction is "single-turn user-agent interaction agentic RL training", as EnvScaler requires the complete instruction be provided in a single turn for **RL**. We acknowledge its improvements in multi-turn, multi-tool interactions (e.g., on $\tau^2$-Bench). We will clarify it in the final version.
> ### Q2 & Originality W1 (Reward Design)
> We would like to clarify that our experiments only show that rule-based rewards is more preferrable than LLM-as-a-judge for ScaleEnv. We do not intend to provide a complete benchmark for various reward designs under different agentic settings, which will be further clarified in our final version.
> ### Q3 & L1 (Large Oracle Model)
> Using the Best-of-16 sampling, the average number of prompt/completion tokens within each task is 8213 ± 1388/16,041 ± 5248. This is on par with the total task construction cost (in our response to **Q3 of Reviewer jgbH**), and the complete statistics will be added in our final version.
> ### Q4 & Soundness W3 & Presentation W1 (Cost & Failure Rate)
> The detailed analysis on the synthesis cost and failure rate can be found in our response to **Q3 of Reviewer jgbH**. Both synthesis cost and failure rate are limited.
> ### Q5 & Soundness W1 (Domain Differences)
> To analyze the structural diversity of generated domains, we consider different graph metrics in the table below. We can see that the **Coefficients of Variation (CV)** for tool count, dependency edges, and density are all very large.
>
> |Metric|Range|CV [2]|
> |-|-|-|
> |Number of nodes (tools)|27–72|21.6|
> |Number of edges|4–70|43.2|
> |Density|0.008–0.043|46.5|
>
> We further provide the **Graphlet Correlation Distance** [3] between the generated domains and domains in VitaBench & $\tau^2$-Bench [*here*](https://anonymous.4open.science/r/scaleenv_rebuttal-69FF/gcd_boxplot_ours_vs_others.png). The average distance to domains in these benchmarks are all **larger than the intra-domain distances within ScaleEnv**, showing that the generated domains **differ significantly in structure** from those in benchmarks.
> ### Soundness W2 & W5 (Comparison of Baselines)
> Additional results and discussions can be found in our response to **Q1 of Reviewer YeKM**, which demonstrate ScaleEnv's  superior generalization performance over these baselines.
> ### Soundness W4 (Training Curve)
> The [*training curve*](https://anonymous.4open.science/r/scaleenv_rebuttal-69FF/reward_curve.png) shows that the training is stable and the reward gradually converges after about 40 steps. The stopping point is selected from the training curve to avoid any issues of depending on a particular stopping point. We will clarify it in our final version.
> ### Soundness W6 (Variance)
> Due to the time limit of rebuttal period, we will include these results in our final version. Please also refer to our response to **W1 of Reviewer jgbH**.
> ### Soundness W7 (Generation Parameters)
> Thank you for raising this question. We have employed a **random sampling strategy** for these generation parameters. For example, for distractor injection density, we uniformly sampled different values for it in range 1-5. As in the table below, training on any single, fixed density level leads to suboptimal performance, while our tasks with mixed levels yields the most robust overall performance. We will include these results in our final version.
>
> |Distractor Injection Density|Cross|Delivery|Instore|OTA|Retail|Airline|Telecom|
> |-|-|-|-|-|-|-|-|
> |1|2|25.5|18.3|6.5|45.0|31.5|28.3|
> |3|2|23.8|18.8|5.8|48.3|34.0|25.2|
> |5|1.3|21.8|20.3|5.3|44.3|33.0|28.3|
> |Ours|3.0|26.3|23.8|7.0|50.9|37.5|27.2|
> ### Presentation W2 (Domain Selection Criteria)
> We generally follow the taxonomy of Table 1 in [1] to ensure the coverage of diverse real-world scenarios. We will clarify it in the final version.
> ### Presentation W3 (Model Role)
> Our workflow follows a **sequential pipeline** with a simple coordination strategy: each agent takes the output of its preceding agent as input. The role assignment is designed to balance cost and performance, and the details can be found [here](https://anonymous.4open.science/r/scaleenv_rebuttal-69FF/models_config.md). We will include these details in our final version.
> ### L2 (Environment Limitations)
> Thank you for your question. ScaleEnv is inherently designed for discrete database states and tool executions. While it covers many existing benchmarks, we will add a separate section to clarify the boundaries of our contribution in our final version.
>
> **References**
>
> [1] The Adoption and Usage of AI Agents: Early Evidence from Perplexity. arXiv preprint 2512.07828
>
> [2] https://en.wikipedia.org/wiki/Coefficient_of_variation
>
> [3] Revealing the Hidden Language of Complex Networks. Scientific Reports

---

> > ### Author Rebuttal · Reviewer_84SD · 2026-04-01
> >
> > Thanks for your kind reply; however, I still have many concerns:
> > 1. I think the investigation on the scope and limitations of the rule-based reward design is interesting for this method. I would like to see evidence for more clearance, for example, whether it would still apply to tasks with less structured or harder-to-formalise success criteria.
> > 2. For a large oracle model, what I am concerned with is the increased budget compared to normal sampling or just simple forward; it's good to see overall consumption, but it seems like it doesn't directly address my problem.
> > 3. While I appreciate the static graph metrics provided (CV, GCD), they don't fully resolve my core concern. Simply having more nodes or a denser graph doesn't prove the underlying reasoning structure is fundamentally different, and you completely omitted the interaction trajectories. Even in a massive 70-node graph, if the agent consistently follows a rigid, linear path, it's functionally still a template variation. To convince me, I need to see an analysis of the dynamic execution paths—such as variance in trajectory lengths, branching, or backtracking—that proves these domains actually require diverse reasoning logic, and should also measure if the success actually comes from such shared patterns with the target benchmark, even though they 'look like' differently.
> > 4. I appreciate the additional baseline comparisons, but the results raise new concerns regarding the actual significance of ScaleEnv's improvements. First, looking at the tau-Bench breakdown, ScaleEnv does not achieve state-of-the-art in any single domain (losing to Simia-Tau in Retail/Airline and EnvScaler in Telecom), acting more as a 'jack of all trades' than a definitive improvement. Second, the performance gains over the strongest baseline (EnvScaler) are extremely marginal: a mere 0.6-point increase on -Bench and a 1.5-point increase on VitaBench. Does this slight bump truly justify the claim of 'superior generalisation'? Furthermore, while you correctly point out Simia-Tau's collapse on VitaBench, ScaleEnv's absolute performance there is still critically low (15.0). Lastly, including TOOLACE with a different base model (LLaMa 3.1) breaks the controlled setting, making it unclear if its poor performance is due to the framework or the underlying LLM. I need a more compelling justification for why these marginal gains constitute a significant contribution.

---

> > > ### Author Response · Authors · 2026-04-04
> > >
> > > Thank you for checking our reply. Here we address your further concerns point by point:
> > >
> > > **Q1 (Reward design)**
> > >
> > > Thank you for your suggestion. We would like to clarify that our work focuses on environment and task synthesis. As such, the comparison of rule-based reward and LLM-as-a-Judge is limited to the scenario of ScaleEnv, where all tasks have structured database and formally verifiable success criteria. Extending such comparison to other types of tasks should be regarded as interesting future work, and we will incorporate such discussion into our final version.
> > >
> > > **Q2 (Oracle model cost)**
> > >
> > > We would like to clarify that the numbers previously reported refers to the cost for **feasibility checking only**. Please also check the table below for more direct comparison. As is mentioned in our previous responses, its cost is generally on par with other stages.
> > >
> > > |Stage|Prompt Tokens|Completion Tokens|
> > > |:---|:---|:---|
> > > |Task & Env Initialization|13,333 ± 2,951|4,748 ± 840|
> > > |Task & Env Completion|22,010 ± 4,348|2,199 ± 601|
> > > |Feasibility Checking|8,213 ± 1,388|16,041 ± 5,248|
> > >
> > > **Q3-1 (Analysis of dynamic execution paths)**
> > >
> > > Thank you for your suggestion. The table below gives statistics of Qwen3-32B trajectories on $\tau^2$-Bench, VitaBench and 3 ScaleEnv domains: project_management (PM), football (FB), and express_logistics (EL). The backtrack/branching rates mean the ratio of backtrack/branching steps in a trajectory.
> > >
> > > | |PM|FB|EL|$\tau^2$|Vita|
> > > |---|---|---|---|---|---|
> > > |Traj length (mean±std)|35±17|25±17|25±21|30±20|47±25|
> > > |% traj w/ backtrack|59|55|41|23|45
> > > |Backtrack rate (%, success/fail)|4/3.2|2.4/6|2.5/2.6|1.3/1.3|1.3/1.9|
> > > |Branching rate (%)|47|56|51|41|4|
> > >
> > > We can see that $\tau^2$-Bench produces trajectories with larger branching rate, while VitaBench have more trajectories with backtrack. As such, simply using domains from a single benchmark easily leads to overfitting. Domains from ScaleEnv produces trajectories with **high length variance, large branching rate and frequent backtracking**, and the backtracking behavior has varying effects: in PM, successful trajectories backtrack *more* than failed ones, whereas in FB the relationship reverses. Such diverse patterns help training models to be generalized to different novel benchmarks.
> > >
> > > **Q3-2 (Shared patterns with target benchmark)**
> > >
> > > While certain universal patterns naturally emerge (e.g., querying by date, filtering by name, chaining IDs), we argue this reflects the inherent structure of real-world agentic tasks rather than benchmark-specific leakage. The key distinction is between superficial template reuse (which we avoid) and learning transferable reasoning primitives (which we intentionally enable). To validate that ScaleEnv achieves the latter, we use a very recent benchmark DeepPlanning [1] to eliminate any possibility of leakage. The table below shows that ScaleEnv-trained model substantially outperforms the base model, indicating that ScaleEnv enables generalization to unseen domains.
> > >
> > > ||Shopping (Match)|Travel (PS)|
> > > |:--- | :---: | :---:|
> > > |Qwen3-8B|34.8|2.1|
> > > |Qwen3-8B + ScaleEnv|47.1|5.8|
> > >
> > > **Q4-1 ($\tau^2$ Performance)**
> > >
> > > We would like to clarify that we do not aim to overtake all baseline methods on all benchmarks, as such can be done simply by synthesizing tasks purely for specific benchmarks. Rather, we emphasize more on its *generalization to unseen domains*, so as to keep competitive performance across diverse domains and avoid possible over-fitting like Simia-Tau. We will incorporate such discussions into our final version.
> > >
> > > **Q4-2 (Comparison with EnvScaler)**
> > >
> > > Despite performance gain, there are two more key advantages of ScaleEnv that seem to be overlooked:
> > > * **Zero-dependency on external models**: existing baseline methods include SFT on trajectories from stronger models
> > > * **Minimal domain priors**: ScaleEnv only require domain keywords as the initial input for domain construction, while all other methods require some additional prior (e.g., specific benchmarks or predefined tasks).
> > >
> > > **Q4-3 (Absolute Performance on VitaBench)**
> > >
> > > Our 15.0 average score on VitaBench (achieved from a small Qwen3-8B model) already surpasses Qwen3-32B (14.8). Also, ScaleEnv can be combined with larger models for better absolute performance: in our Table 1, Qwen3-32B+ScaleEnv achieves an average score of 22.3, closely approaching the performance of large Kimi-K2 model (23.2).
> > >
> > > **Q4-4 (TOOLACE's base model)**
> > >
> > > Since TOOLACE originally selected LLaMa 3.1 as the base model, we followed such settings for exact replication. Here we further provide our self-reporting results using Qwen3-8B, which still falls behind ScaleEnv.
> > > ||**$\tau^2$ Avg**|**Vita Avg**|
> > > |:---|:---:|:---:|
> > > |TOOLACE (LLaMa 3.1-8B)|26.0|1.4|
> > > |Qwen3-8B + TOOLACE Data|24.7|7.4|
> > > |**ScaleEnv (Ours)**|**38.5**|**15.0**|
> > >
> > > **Reference**
> > >
> > > [1] DEEPPLANNING: Benchmarking Long-Horizon Agentic Planning with Verifiable Constraints  arXiv:2601.18137

---

### Official Review · Reviewer_KbBa · 2026-03-13

**Soundness:** 2
**Presentation:** 3
**Significance:** 3
**Originality:** 2
**Overall Recommendation:** 4
**Confidence:** 3

**Summary:**

This paper introduces ScaleEnv, a framework that automatically builds executable tool-use environments, including databases, tools, and verifiable tasks, from only a small amount of domain input. The main idea is to train agents in many diverse synthetic but runnable environments, instead of relying only on static trajectories or text-simulated feedback. Experiments show that this improves generalization on unseen interactive tool-use benchmarks, and the paper argues that environment diversity is especially important for agent training.

**Compliance With Llm Reviewing Policy:**

Affirmed.

**Final Justification:**

The authors' rebuttal have addressed most of my concerns and I have changed my score to 4.

**Key Questions For Authors:**

1. How can the effect of the user simulator be validated? How sensitive are the results to the choice of LLM used as the simulator?

2. What is the cost of each synthetic stage in the pipeline? Could the authors provide a quantitative breakdown of the overall cost?

**Limitations:**

yes

**Strengths And Weaknesses:**

## Strength
1. This paper proposes a new way to build executable tool-use environments from scratch.

2. The whole pipeline is grounded in execution and verification, making the environments and rewards much more reliable.

3. Experiments on two benchmarks show improvemtns on unseen interactive tool-use benchmarks.

## Weakness
1. The paper shows improvements on two benchmarks within its own synthetic training setup, but it does not fully explain which part of the framework is driving the gains. The ablations are still incomplete. For example, it is unclear whether environment diversity or executability is more important.

2. The claim that environment diversity matters more than the number of tasks could be validated more carefully. It would be helpful to compare domain scaling and task scaling directly, to see which one brings more consistent gains.

3. The overall framework is quite complex, and many implementation details are not fully described, which makes reproduction difficult.

---

> ### Author Rebuttal · Authors · 2026-03-31
>
> We sincerely thank you for acknowledging the reliability of our method and experiments, as well as for your constructive feedback.
>
> **W1: Component Contribution: Executability vs. Diversity**
>
> We thank the reviewer for the insightful comment. To clarify the driving forces behind our framework's gains, we address this from two perspectives: **Executability** and **Diversity**. In short, executability is the fundamental prerequisite for successful task construction, while diversity drives the model's generalization capabilities.
>
> **1. Executability (The Foundation)**
> Executability is guaranteed by two core pillars in our framework: *Procedural Testing* and *Graph Expansion*.
>
> *   **Procedural Testing** ensures domain validity. Removing it allows faulty tools to slip into generated domains, causing a significant drop in the success rate of task construction.
> *   **Graph Expansion** ensures interaction completeness and entity consistency. Removing it fundamentally breaks the environment's internal consistency. Without it, the LLM hallucinates environment states, leading to broken tool-execution paths and the steepest drop in the success rate.
>
> |Model/Ablation|Success Rate (%)|$\Delta$|
> |:---|:---:|:---:|
> |**ScaleEnv (Full)**|**66.2**|-|
> |w/o Procedural Testing|39.0|-27.2|
> |w/o Graph Expansion|**27.0**|**-39.2**|
>
> **2. Diversity (The Driver of Generalization)**
> Once executability is secured, environment diversity is what further improves the model's generalization performance. Please refer to our reply to W2 below.
>
> **W2: Domain & Task Scaling**
>
> We greatly appreciate this constructive suggestion. We have systematically analyzed the model performance trained from varying numbers of domains and tasks per domain. The average performance across 4 domains in VitaBench for these models is shown in the table below and will be included in our final version.
>
> ||\# Tasks per-domain = 16|\# Tasks per-domain= 32|\# Tasks per-domain = 64|
> |:---|:---:| :---: | :---: |
> |**\# Domain = 4**|10.2|12.5|12.1|
> |**\# Domain = 8**|13.8|14.1|14.4|
> |**\# Domain = 16**|13.9|14.6|14.5|
>
> We can draw two main conclusions from these results:
> 1.  **Diminishing Returns in Task Scaling:** Increasing the number of tasks per domain from 16 to 32 generally yields noticeable performance improvements. However, further scaling from 32 to 64 tasks provides marginal or no obvious gains, suggesting a saturation point in intra-domain task diversity.
> 2.  **Domain Expansion Outweighs Task Expansion:** Expanding the number of domains yields significantly higher returns than merely increasing the number of tasks within existing domains. For instance, starting from a baseline of Domain=4 and Task=16 (score: 10.2), scaling the tasks by 4x (to Task=64) only improves the performance to 12.1. In contrast, scaling the domains by just 2x (to Domain=8, Task=16) provides a much larger boost to 13.8.
>
> **W3: Implementation Details**
>
> We have elaborated on important implementation details in the paper. Specifically, the processes for code generation validation and database classification comparison validation are detailed in **Section 4.1.1**, while the environment supplementation mechanism is described in **Section 4.2**. Furthermore, we will release all implementation details by open-sourcing our pipeline code.
>
> **Q1: Effect of the User Simulator**
>
> We conducted two experiments that train from Qwen3-8B with two different models as the user simulator. We report their final training reward and Avg@4 on 4 domains in VitaBench. For the evaluation on VitaBench and $\tau^2$Bench, the User Simulator utilizes GPT-4.1, as specified by the benchmarks.
>
> |User Simulator|Train Reward|Avg Vita|
> |:---|:---:|:---:|
> |Qwen3-32B|0.21|13.7|
> |Qwen2.5-72B|0.23|14.5|
>
> **Analysis:**
> As shown in the results, the overall impact of the user simulator's size on the final performance is relatively limited. Therefore, in practice, our primary criteria for selecting a user simulator are simply that it is sufficiently strong and large to follow instructions effectively, while also being capable of being deployed locally for large throughput.
>
> **Q2: Cost of Constructing Environments and Tasks**
>
> To quantify the generation cost, we randomly selected 4 domains (generating 20 tasks per domain) and tracked the resource consumption. The table below reports the per-domain generation metrics.
>
> |Stage|Prompt Tokens|Completion Tokens|Success Rate (%)|Avg. # Retries|
> |:---|:---|:---|:---|:---|
> |Schema Definition (Per Domain)|26,342 ± 1,303|31,326 ± 3,762|100.0|1.00|
> |Schema Implementation (Per Domain)|202,184 ± 18,664|64,719 ± 6,794|89.8 ± 4.6|1.27 ± 0.16|
> |Task & Env Initialization (Per Task)|13,333 ± 2,951|4,748 ± 840|91.7 ± 7.9|1.27 ± 0.27|
> |Task & Env Completion (Per Task)|22,010 ± 4,348|2,199 ± 601|72.2 ± 16.7|1.46 ± 0.03|
>
> *(Note: "Success Rate" here denotes the pass rate at each stage before triggering an automated retry).*

---

> > ### Author Rebuttal · Reviewer_KbBa · 2026-04-03
> >
> > Most of my concerns have been addressed. I will keep my positive score.

---

> > > ### Author Response · Authors · 2026-04-04
> > >
> > > We are delighted that our response has resolved most of your concerns. We sincerely appreciate your encouraging comment about keeping a "positive score" while the current rating is "Weak Reject". **We would be deeply grateful if you could kindly reconsider your score** if our rebuttal has indeed addressed your main concerns. Thank you again for your time, valuable feedback, and willingness to help improve our work. We will ensure all these changes are incorporated into the final version.

---

### Official Review · Reviewer_jgbH · 2026-03-14

**Soundness:** 3
**Presentation:** 2
**Significance:** 3
**Originality:** 2
**Overall Recommendation:** 4
**Confidence:** 3

**Summary:**

This paper proposes ScaleEnv, a framework for synthesizing interactive environments and tasks for tool-use agents. It first constructs executable tool/database graphs through LLM generation plus testing/debugging, and then instantiates solvable tasks with controllable diversity. The paper also uses rule-based rewards defined over final database states instead of LLM-as-a-judge. Experiments train Qwen3-8B and Qwen3-32B on 16 synthetic domains and evaluate on $\tau^2$-Bench and VitaBench. Results suggest that ScaleEnv-trained models outperform baselines across multiple settings, and additional scaling analyses indicate that increasing environment diversity is more beneficial than increasing the number of tasks.

**Compliance With Llm Reviewing Policy:**

Affirmed.

**Final Justification:**

My concerns have been addressed, and I keep my original score.

**Key Questions For Authors:**

1. Does ScaleEnv risk benchmark leakage, for example through implicit overlap in tool structure, task templates, or semantic patterns with VitaBench or $\tau^2$-bench? Although the paper states that test domains are unseen during training, semantic-level similarity could still exist.
2. Which component of ScaleEnv is most responsible for the gains?
3. What is the cost of constructing environments and tasks? It would be valuable to report per-domain generation cost, the number of testing/debugging iterations, retry/failure rates, and the amount of manual intervention required.

**Limitations:**

It would be helpful to analyze the mismatch between synthetic and real environments. If the generated schemas, code, or test cases contain systematic biases or mistakes, these errors may become baked into the training pipeline.

**Strengths And Weaknesses:**

Strengths:
- The paper is clearly written, and the figures are helpful for understanding the pipeline.
- The paper tackles an important bottleneck: scalable, verifiable RL environments for generalist tool-use agents.
- The framework connects databases, tools, execution testing, dependency graphs, and task solvability into a relatively coherent methodology.
- The main experiments cover two model scales, two external benchmarks, multiple domains, and several additional analyses, including ablations on execution verification, reward design, domain scaling, and stability.

Weaknesses:
- In RL and program synthesis settings, variance can be substantial. It would be helpful to report multi-seed results.

---

> ### Author Rebuttal · Authors · 2026-03-31
>
> We sincerely thank the reviewer for recognizing the value of our work and providing constructive feedback. We address your remaining concerns below and will incorporate these discussions into our final version.
>
> **W1: Variance in RL and multi-seed results**
>
> While previous works in this specific area (e.g., EnvScaler [1], Simia [2]) omitted these results due to the prohibitive computational cost of agentic RL training, we agree that reporting multi-seed results is crucial for rigorous evaluation. Due to the time limit of the rebuttal period, we commit to including comprehensive multi-seed evaluations for our main results in the camera-ready version.
>
> **Q1: Benchmark Leakage Risk**
>
> Thank you for raising this. We strictly ensure that **ScaleEnv does not suffer from benchmark leakage**:
> 1. *Tool Structures & Task Templates*. Tools are synthesized entirely from scratch by the LLM without referencing existing codebases. We do not utilize any existing task templates: task instructions are strictly conditioned on the randomly sampled tool chains generated from our pipeline.
> 2. *Semantic Patterns*. As illustrated in Fig. 4 in Appendix A, we deliberately construct domains that are semantically distinct from those in VitaBench and $\tau^2$-bench.
>
> While certain universal algorithmic patterns naturally emerge (e.g., querying by date, filtering by name, chaining IDs), these are inherent mechanics of real-world tasks rather than benchmark-specific leakage. Learning these universal patterns is precisely what allows models trained on ScaleEnv to exhibit strong zero-shot generalization to unseen benchmarks.
>
> **Q2: Component Contribution**
>
> Through our ablation study, we identify **Graph Expansion** (in the task synthesis stage) as the component most responsible for the performance gains.
>
> ScaleEnv relies on two pillars: *Procedural Testing* (to ensure domain validity) and *Graph Expansion* (to ensure interaction completeness and entity consistency). As shown below, while removing *Procedural Testing* introduces faulty tools and significantly drops task construction success rate, removing *Graph Expansion* for task construction fundamentally breaks the environment's internal consistency. Without it, the LLM hallucinates environment states, leading to broken tool-execution paths and the steepest drop in task construction success rate (from 66.2% to 27.0%).
>
> | Model / Ablation | Success Rate (%) | $\Delta$ |
> | :--- | :---: | :---: |
> | **ScaleEnv (Full)** | **66.2** | - |
> | w/o Procedural Testing | 39.0 | - 27.2 |
> | w/o Graph Expansion | **27.0** | **- 39.2** |
>
> **Q3: Cost of Constructing Environments and Tasks**
>
> ScaleEnv is highly cost-effective and, crucially, requires **zero manual intervention**. To quantify the generation cost, we randomly selected 4 domains (generating 20 tasks per domain) and tracked the resource consumption. The table below reports the per-domain generation metrics.
>
> | Stage | Prompt Tokens | Completion Tokens | Success Rate (%) | Avg. # Retries |
> | :--- | :--- | :--- | :--- | :--- |
> | Schema Definition (Per Domain) | 26,342 ± 1,303 | 31,326 ± 3,762 | 100.0 | 1.00 |
> | Schema Implementation (Per Domain) | 202,184 ± 18,664 | 64,719 ± 6,794 | 89.8 ± 4.6 | 1.27 ± 0.16 |
> | Task & Env Initialization (Per Task) | 13,333 ± 2,951 | 4,748 ± 840 | 91.7 ± 7.9 | 1.27 ± 0.27 |
> | Task & Env Completion (Per Task) | 22,010 ± 4,348 | 2,199 ± 601 | 72.2 ± 16.7 | 1.46 ± 0.03 |
>
> *(Note: "Success Rate" here denotes the pass rate at each stage before triggering an automated retry).*
>
> **Limitations: Mismatch Between Synthetic and Real Environments**
>
> We appreciate this insightful point. While ScaleEnv uses *Procedural Testing* as a strict programmatic filter to catch syntax and execution errors, systematic semantic biases (e.g., an LLM's tendency to name variables in specific styles, or generating "happy-path" test cases rather than edge cases) are harder to mechanically filter out. If a model overfits to the "LLM dialect" of the synthetic data, real-world deployment could suffer from distribution shifts. We view the synthesis of mathematically verifiable yet highly adversarial/edge-case environments as a critical next step, and we will prominently discuss this limitation and future direction in our final version.
>
> **Reference**
>
> [1] EnvScaler: Scaling Tool-Interactive Environments for LLM Agent via Programmatic Synthesis. arXiv:2601.05808
>
> [2] Simulating Environments with Reasoning Models for Agent Training. arXiv:2511.01824

---

> > ### Author Rebuttal · Reviewer_jgbH · 2026-04-04
> >
> > The rebuttal has addressed most of my concerns, and I maintain my positive score.

---

> > > ### Author Response · Authors · 2026-04-07
> > >
> > > We are very pleased to hear that our responses have successfully addressed most of your concerns. Thank you very much for continuing to support our paper. We deeply appreciate your constructive feedback, which has been extremely helpful in improving our work.

---

### Official Review · Reviewer_YekM · 2026-03-22

**Soundness:** 3
**Presentation:** 3
**Significance:** 3
**Originality:** 3
**Overall Recommendation:** 5
**Confidence:** 4

**Summary:**

The paper presents an automatic environment construction method: ScaleEnv. The framework uses procedural testing, which guarantees task completeness and solvability via tool dependency graph expansion and executable action verification.

Training on the generated environments shows performance improvement on unseen, multi-turn tool-use benchmarks such as tau-Bench and VitaBench.

**Compliance With Llm Reviewing Policy:**

Affirmed.

**Final Justification:**

The authors provided detailed comparisons to baseline methods in the rebuttal, which answers my major concern.

**Key Questions For Authors:**

(see weakness for details) how's the effectivenss of the proposed method compared to existing environment/tool synthesizing methods?

**Limitations:**

yes

**Strengths And Weaknesses:**

Strength:
- Environment construction is a bottleneck challenge of LLM agent training.
- The construction processes of both tools and tasks are reasonable. The expansion of the environment is rigorous.


Weakness:
- My major concern is that there are no baselines for LLM-based environment construction methods. Although the authors mention a bunch of works for LLM-synthesized environments in the introduction and their weakness, they don't directly reproduce or compare to them in the experiment section. This weakens the claims about the advantage of the proposed method.
- The Tool Schema is purely synthesized by LLMs. In this way, how to guarantee its coverage and naturalness? However, I see this as only a minor weakness because training on such data already gives performance gain on multiple benchmarks.

---

> ### Author Rebuttal · Authors · 2026-03-31
>
> We thank the reviewer for recognizing the value of our work and the constructive feedback. We have carefully addressed your remaining concerns below and will incorporate these additions into the final version.
>
> **W1 & Q1: Comparison with LLM-based Environment Construction Baselines**
>
> We have conducted extensive new experiments comparing **ScaleEnv** against three recent, strong baselines: TOOLACE [1], Simia-Tau [2], and EnvScaler [3]. We evaluated the methods on $\tau^2$-Bench and VitaBench using the Qwen3-8B base model (note: TOOLACE uses Llama 3.1-8B per its original design).
>
> |Method|Retail|Airline|Telecom|**$\tau^2$ Avg**|Cross|Delivery|Instore|OTA|**Vita Avg**|
> |:---|:---:|:---:|:---:|:---:|:---:|:---:|:---:|:---:|:---:|
> |Base Model (Qwen3-8B)|38.4|30.5|21.5| 30.1 | 1.5 | 18.3 | 14.8 | 4.5 | 9.8 |
> | TOOLACE (LLaMa 3.1-8B)| 38.7 | 18.0 | 21.2 | 26.0 | 0.0 | 3.3 | 0.3 | 0.3 | 1.4 |
> | Simia-Tau (Qwen3-8B) | 52.9 | 40.5 | 15.6 | 36.3 | 0.0 | 2.3 | 0.3 | 0.8 | 0.9 |
> | EnvScaler (Qwen3-8B) | 49.6 | 31.5 | 32.7 | 37.9 | 2.5 | 22.8 | 22.3 | 6.3 | 13.5 |
> | **ScaleEnv (Ours)** | **50.9** | **37.5** | **27.2** | **38.5** | **3.0** | **26.3** | **23.8** | **7.0** | **15.0** |
>
> These results demonstrate ScaleEnv's strong overall performance and superior generalization. Notably, while Simia-Tau achieves competitive results on $\tau^2$-Bench, its performance on VitaBench collapses almost to zero. This is due to their different interaction patterns: in $\tau^2$-Bench, *user* determines session termination, while in VitaBench, *agent* determines session termination. As shown in this example [(link)](https://anonymous.4open.science/r/scaleenv_rebuttal-69FF/simia_trajectory.png), instead of ending the conversation, Simia-Tau keeps repeating its responses on VitaBench. This indicates severe overfitting to specific interaction patterns (relying the user to end the conversation). In contrast, ScaleEnv improves generalization across unseen domains without fixed behavior collapse, validating the importance of our diverse environment generation approach.
>
> **Methodological Advantages.**
> To clarify *why* ScaleEnv generalizes better, we summarize the characteristics of these baselines:
>
> | Method | Env Construction | Training Paradigm | Teacher Model Dependency | Prior Dependency |
> |:---|:---|:---|:---|:---|
> | TOOLACE | LLM-simulated | SFT | High (GPT-4o) | Pretraining Data |
> | Simia | LLM-simulated | SFT + RL | High (GPT-5, GPT-4o-mini) | Specific Benchmarks |
> | EnvScaler| LLM-synthesized | SFT + RL | High (Strong Reasoning Models) | Predefined Task Sets|
> | **ScaleEnv**| **LLM-synthesized**| **Pure RL** | **None (Teacher-free)** | **Domain Keywords** |
>
> Unlike previous methods, ScaleEnv (1) completely eliminates the bottleneck of distilling from expensive, proprietary teacher models by utilizing agent self-exploration via RL, and (2) requires minimal prior dependency (only lightweight domain keywords) compared to rigid predefined task sets or reference benchmarks, making it inherently more scalable.
>
> **W2: Guaranteeing Coverage and Naturalness of Synthesized Tool Schemas**
>
> Thank you for raising this insightful question. To explicitly evaluate the coverage and naturalness of our synthesized tool schemas, we use the **Tool Dependency Graph** [(link)](https://anonymous.4open.science/r/scaleenv_rebuttal-69FF/ScaleEnvGraph/email_management.png) mapping the dependencies between the tools' input parameters, output parameters, and database fields. From this graph,
> *   **Guaranteeing Naturalness via Tool Walks:** To ensure logical consistency (naturalness), the algorithm performs graph traversals (tool walks) on the Tool Dependency Graph to actively check for isolated nodes (e.g., tools with unfulfillable inputs or unutilized outputs). Once such anomalies are detected, the system resolve them by either adopting a discard strategy to reject the invalid schema, or employing an LLM supplementation strategy to refine it. This structural constraint forces the generated schemas to mimic natural, usable API designs.
> *   **Guaranteeing Coverage via Database Field Mapping:** To ensure completeness (coverage), the algorithm systematically checks whether every field in the target domain database is associated with at least one tool (either for querying or modification). If any unmapped fields are identified, the system addresses this gap by either discarding the incomplete schema or utilizing the LLM supplementation strategy to generate the missing tools.
>
> Comprehensive graph shows that ScaleEnv's generated schemas maintain dense connectivity and near-total database coverage.
>
> **Reference**
>
> [1] Toolace: Winning the points of llm function calling. ICLR 2025
>
> [2] Simulating Environments with Reasoning Models for Agent Training. arXiv:2511.01824
>
> [3] EnvScaler: Scaling Tool-Interactive Environments for LLM Agent via Programmatic Synthesis. arXiv:2601.05808

---

> > ### Author Rebuttal · Reviewer_YekM · 2026-04-01
> >
> > Thanks for providing additional results on tau-bench. It's nice to see that the proposed method outperforms EnvScaler, the strongest baseline, on most domains and on the avg score. I've raised my score.

---

> > > ### Author Response · Authors · 2026-04-04
> > >
> > > We are glad to hear that you found these additional results helpful in strengthening the empirical support for our method, which will all be incorporated into our final version. Thank you again for your time and valuable feedback throughout the review process.

---

### Decision · Program_Chairs · 2026-04-30

**Decision:**

Accept (regular)

**Comment:**

This paper presents ScaleEnv, a framework for constructing fully interactive environments and verifiable tasks from scratch to train generalist tool-use agents via reinforcement learning. The reviewers recognize that this addresses a critical bottleneck: the scarcity of scalable, verifiable RL environments for tool-use agents. The pipeline is well-structured, connecting databases, tools, execution testing, dependency graphs, and task solvability into a coherent methodology grounded in execution and verification. A notable design feature is that the framework is fully teacher-free, requiring zero dependency on external proprietary models and only domain keywords as input.

The main concerns centered on the lack of direct comparisons with other synthetic environment methods and unclear structural diversity of the generated domains. During the rebuttal, the authors provided comparisons with TOOLACE, Simia-Tau, and EnvScaler showing favorable results, trajectory dynamics analysis demonstrating diverse branching and backtracking patterns, training curves showing convergence, and generalization on the DeepPlanning benchmark. Three of four reviewers confirmed full resolution of concerns. The remaining reviewer maintained a borderline-reject score, primarily citing marginal gains over EnvScaler; however, EnvScaler is concurrent work (arXiv 2601.05808) and per ICML policy should not negatively impact evaluation. Multi-seed variance results were not provided during rebuttal due to time constraints and should be included in the camera-ready version.